# Beyond RAG vs. Long-Context: Learning Distraction-Aware Retrieval for Efficient Knowledge Grounding

**Seongwoong Shim**[1,*]     **Myunsoo Kim**[1,*]     **Jae Hyeon Cho**[2]     **Byung-Jun Lee**[1,†]

[1]Korea University, Decision Making Lab    [2]LG CNS

[1]{ssw030830, m970326, byungjunlee}@korea.ac.kr   [2]jh.cho@lgcns.com

## Abstract

Retrieval-Augmented Generation (RAG) is a framework for grounding Large Language Models (LLMs) in external, up-to-date information. However, recent advancements in context window size allow LLMs to process inputs of up to 128K tokens or more, offering an alternative strategy: supplying the full document context directly to the model, rather than relying on RAG to retrieve a subset of contexts. Nevertheless, this emerging alternative strategy has notable limitations: (i) it is token-inefficient to handle large and potentially redundant contexts; (ii) it exacerbates the 'lost in the middle' phenomenon; and (iii) under limited model capacity, it amplifies distraction, ultimately degrading LLM output quality. In this paper, we propose LDAR (Learning Distraction-Aware Retrieval), an adaptive retriever that learns to retrieve contexts in a way that mitigates interference from distracting passages, thereby achieving significantly higher performance with reduced token usage compared to long-context approaches. Extensive experiments across diverse LLM architectures and six knowledge-intensive benchmarks demonstrate the effectiveness and robustness of our approach, highlighting the importance of balancing the trade-off between information coverage and distraction.

## 1 Introduction

Despite the remarkable progress of Large Language Models (LLMs), they continue to exhibit factual errors (Wei et al., 2024; Lv et al., 2024; Li et al., 2024a), and their knowledge remains limited to the static dataset on which they were trained. To address these limitations, Retrieval-Augmented Generation (RAG) has been proposed, enabling models to ground their outputs in external, up-to-date information, thereby enhancing both accuracy and relevance in knowledge-intensive tasks (Zhang et al., 2024a; Xu et al., 2024). In practice, RAG retrieves a small set of the most relevant passages from an external corpus to ground the LLM's generation process.

Recent advancements have substantially increased the context length of LLMs, with some models now supporting inputs of up to 128K tokens or more (e.g., GPT-4o (OpenAI, 2024), Gemini 2.5 (Comanici et al., 2025), Qwen 2.5 (Yang et al., 2025)). This capability offers an alternative strategy for grounding model outputs: supplying the full document context directly to the model, rather than relying on RAG

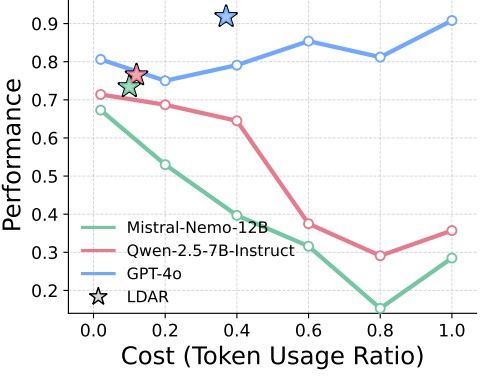

Figure 1: Performance of LLMs across token usage ratios. Higher ratio corresponds to retrieving more passages. Lines indicate performance when retrieving top-similarity passages within a fixed token usage ratio (1.0 = full context). ☆ marks the performance of LDAR optimized for each LLM, illustrating its ability to strike a balance between information coverage and distraction that surpasses all fixed token usage baselines.

---

[*]Equal contribution.

[†]Corresponding author.

to supply only a subset of them. With sufficient capacity, LLMs can selectively attend to salient information while disregarding irrelevant content, thereby reducing reliance on explicit retrieval mechanisms. Indeed, empirical evidence from emerging benchmarks indicates that providing LLMs with full long-contexts frequently outperforms RAG-based approaches (Li et al., 2024b; Wang et al., 2024b; Li et al., 2025a). Nevertheless, this long-context approach has its own drawbacks, as it can be token-inefficient to process large, potentially redundant contexts. Moreover, long-context approaches are prone to the 'lost in the middle' phenomenon, where the model struggles to recall information presented in the middle of a long input sequence (Liu et al., 2023). When model capacity is limited, supplying the full context may further introduce distraction, thereby degrading answer quality (Li et al., 2025a). These challenges highlight the need for approaches that integrate the advantages of both paradigms—approaching the performance of long-context approaches while maintaining the token efficiency of RAG.

In this paper, we demonstrate that both open and closed-source LLMs can still fail to answer questions even when the gold passage is retrieved, due to interference from additionally retrieved passages (i.e., distracting passages) (Shi et al., 2023a; Cuconasu et al., 2024; Amiraz et al., 2025). However, retrieving passages to minimize such distraction remains a non-trivial challenge, as the optimal strategy depends not only on the capacity of the target LLM, but also on the combinatorial interactions among the retrieved passages. To address this challenge, we introduce LDAR (Learning Distraction-Aware Retrieval), a retriever that learns to select passages to minimize potential interference from distracting passages in accordance with the capacity of the LLM, thereby achieving significantly better performance and lower token usage compared to long-context approaches. In summary, our contributions are as follows:

1. Unlike previous heuristic-based methods, we propose a learning-based retrieval strategy framework that adaptively balances information coverage and distraction in accordance with the capacity of the LLM, achieving better performance with significantly reduced token usage compared to the long-context approach.

2. We empirically demonstrate that retrieving passages in bands (i.e., selecting from contiguous ranges along the similarity-ranked list) is critical for learning a distraction-aware retrieval strategy. The banded retrieval strategy provides a form of abstraction that improves generalization and prevents the retriever from converging to suboptimal solutions.

3. We validate our approach across diverse LLM architectures (both open and closed-source) and six knowledge-intensive benchmarks, demonstrating both the effectiveness and robustness of the proposed retrieval strategy. Our code is at https://github.com/ku-dmlab/LDAR.

## 2 Related Works

**RAG vs. Long-context LLM** Numerous studies have examined the comparative performance of RAG versus LLMs provided with the entire input context. While some works report that RAG outperforms long-context approaches (Xu et al., 2023b), other works demonstrate the opposite trend, with long-context models surpassing RAG-based methods (Li et al., 2024b). Li et al. (2025a) demonstrates that this divergence in findings largely stems from the capacity of LLMs used to evaluate the results. Open-source LLMs typically exhibit limited capacity for processing long contexts and therefore benefit substantially from retrieval mechanisms. On the other hand, closed-source LLMs often possess stronger long-context capabilities and consequently achieve higher performance when given full-context inputs (Li et al., 2025a). These findings suggest that RAG works as a stopgap technique to boost models that otherwise struggle with long sequences (Bai et al., 2023; Li et al., 2025a). Furthermore, several studies highlight that increasing the number of retrieved text chunks yields an inverted-U pattern: performance initially improves but eventually declines as the model becomes distracted by irrelevant or misleading passages (Jin et al., 2024; Leng et al., 2024). This observation underscores the need for retrieval strategies that balance information coverage against the risk of distraction, thereby optimizing the trade-off in passage selection.

**Bridging the Gap between Retriever and LLM** Our method can also be interpreted as bridging the gap between the retriever and the LLM. Since retrievers and LLMs are pretrained under distinct training objectives and architectures, a preference gap naturally arises between them (Ke et al., 2024; Ye et al., 2024). The passages retrieved by the retriever can even distract the LLM during answer generation, thereby degrading downstream performance (Shi et al., 2023a; Cuconasu et al., 2024;

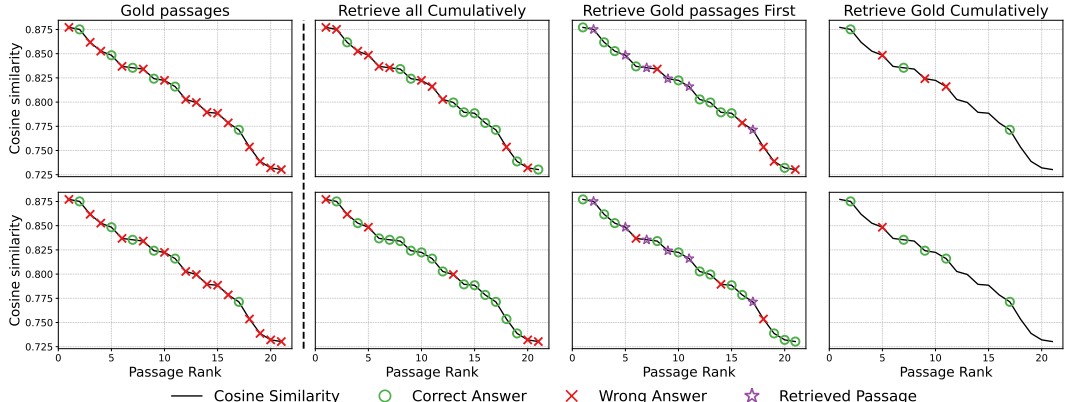

Figure 2: Visualization of different retrieval strategies and their impact on performance. A green circle (○) indicates that retrieving the passage yields a correct answer, a red cross (✗) indicates retrieving the passage yields a wrong answer, and a purple star (☆) denotes a passage that has already been incorporated into the retrieved passage set. The black curve represents the cosine similarity between the query and passages. The top row reports results for an open-source model (Llama-3.1-8B), while the bottom row shows results for a closed-source model (GPT-4o) on a reasoning task (Li et al., 2025a).

Amiraz et al., 2025). Prior work has attempted to mitigate this gap by fine-tuning the LLM (Izacard & Grave, 2020; de Jong et al., 2023), fine-tuning the retriever (Shi et al., 2023b; Xu et al., 2023a), jointly fine-tuning both components (Izacard et al., 2022; Lewis et al., 2020), or training a module that bridges the gap (Ke et al., 2024; Ye et al., 2024). Whereas bridge modules identify relevant passages based on textual information within the top-$k$ candidates retrieved by cosine similarity, our method instead targets the retrieval stage itself. Specifically, we aim to retrieve sets of passages that minimize distraction under a fixed pretrained retriever and LLM, relying solely on the similarity distribution between the query and passages.

## 3 MOTIVATION

Although prior works have highlighted the detrimental impact of distracting passages on retrieval performance (Jin et al., 2024; Leng et al., 2024), relatively little attention has been paid to retrieval strategies that explicitly mitigate such influence. The columns in Figure 2 illustrate retrieval strategies commonly adopted in practice: (1) gold passages that lead to correct answer when individually retrieved; (2) retrieving all passages cumulatively from top to bottom, which corresponds to the conventional top-$k$ similarity-based retrieval approach (Lewis et al., 2020; Karpukhin et al., 2020); (3) retrieving all gold passages first (☆) followed by retrieving additional passages, which corresponds to reranking the retrieved top-$k$ passages to prioritize relevance (Nogueira & Cho, 2019; Nogueira et al., 2020; Glass et al., 2022); (4) retrieving gold passages cumulatively from top to bottom, which resembles the successful outcome of the hybrid strategy: first selecting the top-$k$ passages by similarity and then applying a relevance-based top-$n$ selection to retain only the gold passages (Asai et al., 2024; Ke et al., 2024; Lee et al., 2025).

As shown in (2), top-$k$ retrieval is susceptible to distracting passages. Even when individually correct passages are included, their joint presence with distracting passages can result in incorrect answers. Note that retrieving all available passages is effectively equivalent to the long-context approach setting, which is likewise prone to errors. In (3), reranking approaches that place highly relevant passages at the front of the retrieved passages still fail when distracting passages are present, as their inclusion can override or obscure the signal from the relevant ones. Finally, (4) shows that even when the hybrid strategy, which combines similarity-based top-$k$ retrieval with relevance-based top-$n$ selection, successfully retains only the gold passages, their joint retrieval can still lead to incorrect answers. Counterintuitively, even when all passages individually lead to the correct answer, their collective inclusion can complicate the reasoning process and ultimately cause the model to generate incorrect outputs. Based on these observations, we define ***distracting passages*** as passages that misguide the LLM in generating the correct answer, irrespective of whether they lead to correct

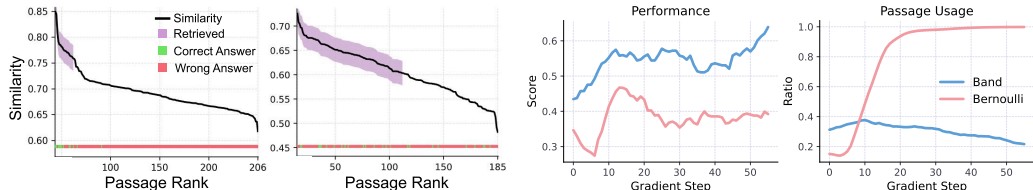

Figure 3: **(Left)** Visualization of passages retrieved by $\pi_\theta$ based on the similarity distribution between queries and passages, with retrieved passages marked in green if they contain the correct answer and in red otherwise. **(Right)** Comparison of performance and passage usage ratio across Bernoulli- and band-based retrieval strategies across gradient steps.

answer when individually retrieved. Furthermore, the differing outcomes in Figure 2 (top vs. bottom) demonstrate that retrieval effectiveness is strongly tied to the capacity of the underlying LLM.

These findings underscore the inherent difficulty of reliably retrieving passages that yield correct answers, motivating the need for retrieval strategies that explicitly account for distracting effects in relation to model capacity. Note that a brute-force remedy to minimize distraction would be to employ a high-capacity LLM to exhaustively read and align every passage to the query. However, this approach is prohibitively expensive, as inference cost scales with the number of passages, making it infeasible in practice (especially in the long-context setting). To address this challenge, we propose a lightweight adaptive retriever framework that learns to minimize distraction by selecting effective passage sets in accordance with the long-context capability of LLMs. This method relies solely on the cosine similarity distribution between queries and passages, guided by evaluation signals to learn an effective balance between information coverage and distraction.

## 4 MAIN METHOD

RAG employs a pretrained embedding model $f_\phi$ that maps a query $q$ and passages $\{p_i\}_{i=1}^N$ into a shared vector space $\mathbb{R}^d$, retrieving the top-$k$ passages ranked by semantic similarity. Let $s_i$ denote the similarity (e.g., dot product or cosine similarity) between the query and the $i$-th passage,

$$\exists\,\sigma \text{ s.t. } s_{\sigma(1)} \leq s_{\sigma(2)} \leq \ldots \leq s_{\sigma(N)}, \quad \mathcal{R} = \{p_{\sigma(N-k+1)}, \ldots, p_{\sigma(N)}\}. \tag{1}$$

Retrieved passages $\mathcal{R}$ with higher similarity scores are semantically closer to the query, as the embedding space is trained via contrastive objectives that pull matched query–passage pairs together while pushing apart mismatched pairs (Izacard et al., 2021; Li et al., 2023; Zhang et al., 2024b). This dense retrieval approach has remained the dominant strategy due to its effectiveness in retrieving relevant passages at low computational cost, spanning from early RAG models to recent applications (Lewis et al., 2020; Tang & Yang, 2024; Asai et al., 2024; Li et al., 2025b).

In this section, we present our lightweight retriever $\pi_\theta$ that learns to select passages to minimize potential interference from distracting passages, solely based on the similarity between the query and passages. As discussed in Section 3, employing a high-capacity LLM to exhaustively align all passages with the query is impractical, particularly in long-context settings. To ensure our approach scales to large-scale retrieval scenarios, we deliberately restrict $\pi_\theta$ from accessing textual information. Moreover, our approach avoids the expense of fine-tuning the large pretrained LLM and embedding model by training only a lightweight neural network to reduce distraction, while keeping the larger components fixed.

Our retriever $\pi_\theta$ operates on the cosine similarity distribution and selects a dynamic set of passages from a contiguous quantile interval $q_L, q_U \subset [0, 1]$. When the retriever $\pi_\theta$ determines that information coverage should be prioritized at the risk of increased distraction, it retrieves passages from a wide quantile interval. On the contrary, if the risk of having distraction is higher, $\pi_\theta$ retrieves passages from a narrow quantile interval, minimizing the risk of having distraction. Figure 3 (left) illustrates that optimized $\pi_\theta$ adapts its retrieval strategy based on the similarity distribution between the query and passages. If passages with high semantic similarity exist, $\pi_\theta$ tends to focus narrowly on that region. In contrast, when no passages exhibit strong semantic similarity, $\pi_\theta$ expands the retrieval range to ensure broader information coverage, even at the cost of incorporating more potential distracting passages.

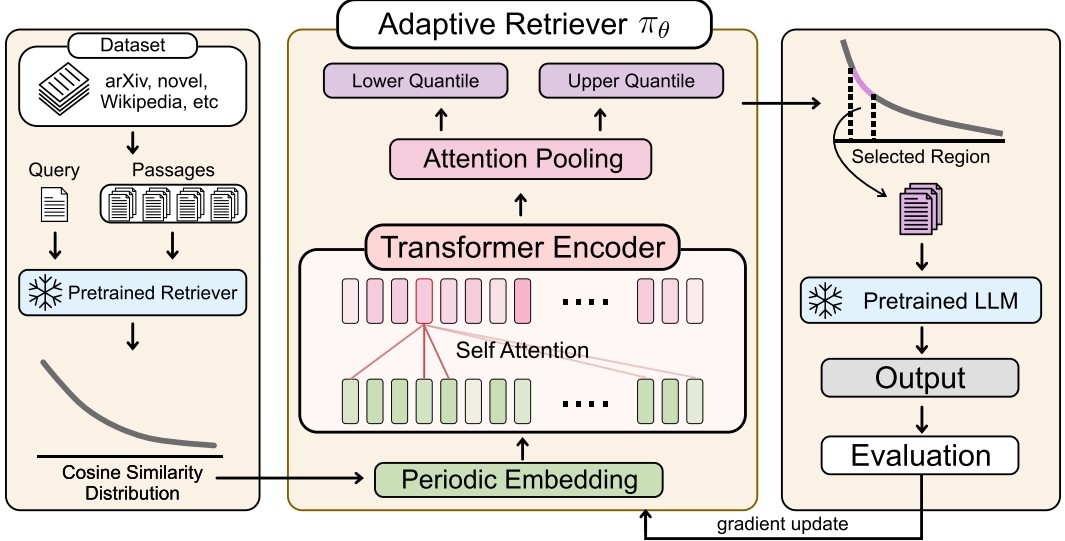

Figure 4: Overview of LDAR, a learning-based retrieval strategy that adapts to each LLM by balancing information coverage and distraction. Given a query, a fixed pretrained retriever computes cosine similarity scores between the query and passages. Then periodic embeddings encode each score into a token, followed by a Transformer encoder that processes the tokenized similarity distribution. The encoder representations are aggregated via attention pooling, after which two output heads predict the lower and upper quantiles that define the similarity interval used for retrieval. The selected passages are passed to a pretrained LLM for prediction, and the evaluation signal is used to update the adaptive retriever through gradient-based learning.

Notably, this behavior also depends on the capability of the pretrained LLM: models with stronger long-context processing generally exhibit lower susceptibility to distraction compared to those with limited long-context capability (Li et al., 2025a). Later in experiments, we demonstrate that the LDAR framework adaptively retrieves fewer passages for open-source models compared to closed-source models on the same task, indicating that our framework aligns retrieval strategies with the long-context capability of the underlying LLM.

## 4.1 Designing the Adaptive Retriever

We provide the adaptive retriever $\pi_\theta$ with a cosine similarity vector $s \in \mathbb{R}^N$ between the query and the $N$ passages, computed by a pretrained embedding model $f_\phi$:

$$s_i := \frac{f_\phi(q)^\top f_\phi(p_i)}{\|f_\phi(q)\| \, \|f_\phi(p_i)\|}, \quad i = 1, \dots, N. \tag{2}$$

Since the number of passages associated with each query may differ, the dimensionality of $s \in \mathbb{R}^N$ is not fixed and varies across queries. To accommodate this variability, we employ a bidirectional self-attention Transformer that maps the token embedding of each similarity score $s_i$ to a contextualized representation. An attention-pooling layer then aggregates these token-level representations into a global summary vector, which is fed to output heads that predict the parameters $(\alpha_L, \beta_L)$ and $(\alpha_U, \beta_U)$ of two Beta distributions. The lower and upper quantiles $q_L$ and $q_U$ are then sampled from these distributions respectively, and the resulting band $\{q_L, q_U\} \subset [0, 1]$ is used to select the passages from the similarity distribution.

Intuitively, allowing the adaptive retriever $\pi_\theta$ to select passages via independent Bernoulli sampling for each candidate is also a valid strategy. However, as shown in Figure 3 (right), the Bernoulli-based variant of LDAR fails to identify a balanced trade-off between the RAG and long-context approach. This limitation arises from its need to explore the entire combinatorial subset selection space, which impedes generalization and ultimately causes convergence to a local optimum (corresponding to the long-context approach in this case). In contrast, band-based retrieval reduces the effective search space from combinatorial subset selection to a low-dimensional and smooth control space, yielding

---

**Algorithm 1** Distraction-Aware Adaptive Retrieval

---

**Require:** Instances $\mathcal{D} = \{(q_m, P_m, y_m)\}_{m=1}^M$, Embedding model $f_\phi$, Adaptive Retriever $\pi_\theta$

1: **for** $m$-th query $q_m$ and passages $\{p_{m,i}\}_{i=1}^N$ in $\mathcal{D}$ **do**
2: $\quad s_i \leftarrow \frac{f_\phi(q_m)^\top f_\phi(p_{m,i})}{\|f_\phi(q_m)\| \|f_\phi(p_{m,i})\|}$ $\quad$ for $i = 1, \cdots, N$
3: $\quad (q_L, q_U) \sim \pi_\theta(\cdot \mid s)$
4: $\quad \ell \leftarrow \max(1, \lfloor N \cdot q_L \rfloor)$
5: $\quad u \leftarrow \max(\ell, \lfloor N \cdot q_U \rfloor)$
6: $\quad \sigma \leftarrow \mathrm{argsort}(s)$ s.t. $s_{\sigma(1)} \leq s_{\sigma(2)} \leq \cdots \leq s_{\sigma(N)}$
7: $\quad \mathcal{R}_m \leftarrow \{p_{\sigma(\ell)}, p_{\sigma(\ell+1)}, \ldots, p_{\sigma(u)}\}$
8: **end for**
9: **return** $\{\mathcal{R}_m\}_{m=1}^M$

---

a temporally abstract retrieval strategy that enables more sample-efficient credit assignment and promotes better exploration (Baranes & Oudeyer, 2013; Machado et al., 2023; Kim et al., 2025). Ultimately, our band-based retrieval strategy allows $\pi_\theta$ to achieve an effective trade-off between information coverage and distraction, yielding a higher score while maintaining a lower passage-retrieval ratio. (see Figure 3 (right)). We provide an illustration of our adaptive retrieval process in Figure 4, with corresponding pseudo-code in Algorithm 1.

## 4.2 OPTIMIZING THE ADAPTIVE RETRIEVER

The main goal of $\pi_\theta$ is to retrieve a set of passages that maximizes the likelihood of the pretrained LLM producing the correct answer to a given query. To this end, we formulate the objective as maximizing the prediction accuracy of the LLM conditioned on the passage set retrieved by $\pi_\theta$:

$$\max_\theta \ J(\theta) = \mathbb{E}_{(q,P,y)\sim\mathcal{D}, \mathcal{R}\sim\pi_\theta(\cdot|s)} \left[ r_\psi(q, \mathcal{R}, y) \right], \text{ where } r_\psi(q, \mathcal{R}, y) := \mathbb{1}_{\mathrm{corr}} \left( F_\psi(q, \mathcal{R}), y \right). \quad (3)$$

Here, $\mathcal{D}$ denotes the dataset with each instance comprising a query $q$, a candidate passage pool $P$, and a ground-truth answer $y$. $\mathcal{R}$ denotes the set of passages retrieved by $\pi_\theta$ given the similarity scores $s$, and $\mathbb{1}_{\mathrm{corr}}$ is an indicator function that evaluates whether the output of the LLM $F_\psi(q, \mathcal{R})$ matches ground-truth answer $y$.

By applying the likelihood ratio gradient with log-derivative trick (Sutton et al., 1999), we can update $\theta$ at $k$-th gradient update step as:

$$\theta_{k+1} = \theta_k + \gamma \cdot r_\psi(q, \mathcal{R}, y) \cdot \nabla_{\theta_k} \log \pi_{\theta_k}(\cdot|s), \quad (4)$$

where $\gamma$ denotes the step size. Through this optimization, $\pi_\theta$ learns a distraction-aware retrieval strategy that reduces the likelihood of distracting passages interfering with the prediction LLM.

## 5 EXPERIMENTS

### 5.1 TASKS AND DATASETS

Six datasets encompassing diverse tasks and contexts are used to evaluate LDAR with other baselines. Each dataset is partitioned into training and test sets using an 8:2 split ratio, and performance is assessed on the test set. Both the training and test sets are available in our GitHub repository.

**Location**, **Reasoning**, **Comparison**, **Hallucination** tasks are from the LaRA benchmark (Li et al., 2025a), which is delicately designed to compare the performance between RAG and long-context approach. Notably, these tasks include contexts approaching the maximum supported length of mainstream commercial and open-weight models (128K tokens), thereby providing a rigorous evaluation of the long-context capabilities of LLMs. In LaRA, contexts are drawn from novels, financial statements, and academic papers with entity replacement (Li et al., 2020; Zhang et al., 2024c) to mitigate the risk of data leakage. The Location task evaluates an LLM's ability to identify precise information based on the provided context. The Reasoning task examines the model's capacity for logical inference, deduction, or computation within the given context. The Comparison task assesses whether the model can integrate and contrast information across multiple parts of the provided context. The

Table 1: Comparison of retrieval strategies across context lengths and task types. Each cell reports the average score across LLMs with standard error. White background indicates the average over open-source LLMs and brown background indicates the average over closed-source LLMs . Numbers in parentheses denote the token-usage ratio relative to LC. The best performing strategy for each task is highlighted in bold. The open-source and closed-source models used to compute the scores are introduced in Section 5.2. The full results are available in Appendix D.11.

| Method | Location | | Reasoning | | Comparison | | Hallucination | | Overall | |
|---|---|---|---|---|---|---|---|---|---|---|
| *Context Length 32k* | | | | | | | | | | |
| Top-1 | $52.7_{\pm 0.5}$ (0.019) | $56.6_{\pm 0.9}$ (0.019) | $23.3_{\pm 3.2}$ (0.019) | $37.1_{\pm 2.2}$ (0.019) | $20.6_{\pm 2.2}$ (0.018) | $33.0_{\pm 3.6}$ (0.018) | $\mathbf{86.0}_{\pm 3.1}$ (0.019) | $\mathbf{89.4}_{\pm 2.8}$ (0.019) | $45.65_{\pm 2.3}$ (0.018) | $54.02_{\pm 2.4}$ (0.018) |
| Top-5 | $66.7_{\pm 1.2}$ (0.095) | $78.0_{\pm 0.5}$ (0.095) | $38.2_{\pm 4.8}$ (0.097) | $61.6_{\pm 4.8}$ (0.097) | $47.6_{\pm 2.7}$ (0.091) | $62.8_{\pm 3.8}$ (0.091) | $82.5_{\pm 3.4}$ (0.095) | $84.3_{\pm 3.3}$ (0.095) | $58.75_{\pm 3.0}$ (0.094) | $71.67_{\pm 3.1}$ (0.094) |
| Top-10 | $75.3_{\pm 1.6}$ (0.190) | $83.4_{\pm 1.1}$ (0.190) | $41.6_{\pm 4.0}$ (0.194) | $59.5_{\pm 3.0}$ (0.194) | $51.5_{\pm 5.9}$ (0.182) | $65.2_{\pm 3.6}$ (0.182) | $79.1_{\pm 6.1}$ (0.190) | $80.8_{\pm 4.3}$ (0.190) | $61.87_{\pm 4.4}$ (0.189) | $72.22_{\pm 3.0}$ (0.189) |
| Top-25 | $78.1_{\pm 3.0}$ (0.474) | $87.4_{\pm 1.2}$ (0.474) | $39.9_{\pm 4.0}$ (0.486) | $61.6_{\pm 0.9}$ (0.486) | $50.2_{\pm 6.9}$ (0.457) | $70.9_{\pm 3.7}$ (0.457) | $74.4_{\pm 8.6}$ (0.476) | $77.0_{\pm 4.8}$ (0.476) | $60.65_{\pm 5.6}$ (0.473) | $74.22_{\pm 2.7}$ (0.473) |
| Top-50 | $78.1_{\pm 2.8}$ (0.866) | $87.0_{\pm 2.5}$ (0.866) | $37.4_{\pm 4.0}$ (0.897) | $63.8_{\pm 2.6}$ (0.897) | $49.6_{\pm 4.2}$ (0.853) | $69.3_{\pm 5.4}$ (0.853) | $72.1_{\pm 10.6}$ (0.862) | $74.7_{\pm 5.0}$ (0.862) | $59.30_{\pm 5.4}$ (0.869) | $73.70_{\pm 3.9}$ (0.869) |
| LC | $80.3_{\pm 1.9}$ (1.000) | $87.4_{\pm 0.7}$ (1.000) | $36.9_{\pm 4.0}$ (1.000) | $62.2_{\pm 3.2}$ (1.000) | $47.7_{\pm 5.3}$ (1.000) | $73.5_{\pm 4.0}$ (1.000) | $69.6_{\pm 8.4}$ (1.000) | $73.0_{\pm 7.3}$ (1.000) | $58.62_{\pm 4.9}$ (1.000) | $74.00_{\pm 3.8}$ (1.000) |
| RAG | $76.3_{\pm 1.7}$ (0.095) | $82.5_{\pm 0.5}$ (0.095) | $43.7_{\pm 3.2}$ (0.097) | $59.0_{\pm 1.3}$ (0.097) | $47.0_{\pm 6.5}$ (0.091) | $61.2_{\pm 5.7}$ (0.091) | $81.5_{\pm 3.7}$ (0.095) | $79.8_{\pm 4.7}$ (0.095) | $62.12_{\pm 3.8}$ (0.094) | $70.62_{\pm 3.1}$ (0.094) |
| Self-Route | $80.6_{\pm 1.7}$ (0.255) | $89.6_{\pm 0.8}$ (0.295) | $40.3_{\pm 5.0}$ (0.258) | $62.7_{\pm 4.1}$ (0.232) | $47.0_{\pm 4.6}$ (0.244) | $67.6_{\pm 2.9}$ (0.312) | $69.9_{\pm 8.2}$ (0.949) | $76.0_{\pm 5.2}$ (0.967) | $59.45_{\pm 4.9}$ (0.426) | $73.97_{\pm 3.3}$ (0.451) |
| Adaptive-$k$ | $61.0_{\pm 2.2}$ (0.395) | $71.4_{\pm 2.2}$ (0.395) | $25.4_{\pm 0.4}$ (0.362) | $51.5_{\pm 3.5}$ (0.362) | $43.1_{\pm 3.0}$ (0.385) | $59.6_{\pm 3.8}$ (0.385) | $77.3_{\pm 7.1}$ (0.479) | $82.2_{\pm 4.4}$ (0.479) | $51.70_{\pm 3.2}$ (0.405) | $66.17_{\pm 3.5}$ (0.405) |
| BGM | $78.8_{\pm 0.9}$ (0.048) | $82.6_{\pm 0.8}$ (0.057) | $46.5_{\pm 4.0}$ (0.067) | $59.0_{\pm 1.6}$ (0.074) | $50.1_{\pm 1.6}$ (0.066) | $61.2_{\pm 6.9}$ (0.064) | $75.4_{\pm 4.7}$ (0.049) | $75.7_{\pm 2.8}$ (0.045) | $62.70_{\pm 2.8}$ (0.057) | $69.63_{\pm 3.0}$ (0.060) |
| RankZephyr | $72.4_{\pm 1.2}$ (0.095) | $82.1_{\pm 1.2}$ (0.095) | $36.9_{\pm 4.7}$ (0.097) | $59.5_{\pm 2.7}$ (0.097) | $44.4_{\pm 4.3}$ (0.091) | $62.1_{\pm 1.8}$ (0.091) | $80.5_{\pm 4.4}$ (0.095) | $85.2_{\pm 2.4}$ (0.095) | $58.55_{\pm 3.7}$ (0.094) | $72.2_{\pm 2.8}$ (0.094) |
| LDAR | $\mathbf{87.7}_{\pm 1.4}$ (0.478) | $\mathbf{91.9}_{\pm 1.2}$ (0.628) | $\mathbf{52.7}_{\pm 4.7}$ (0.400) | $\mathbf{70.1}_{\pm 0.9}$ (0.636) | $\mathbf{63.1}_{\pm 2.6}$ (0.518) | $\mathbf{78.9}_{\pm 4.0}$ (0.619) | $76.5_{\pm 6.9}$ (0.474) | $76.8_{\pm 5.8}$ (0.633) | $\mathbf{70.00}_{\pm 3.9}$ (0.467) | $\mathbf{79.42}_{\pm 3.0}$ (0.629) |
| *Context Length 128k* | | | | | | | | | | |
| Top-1 | $31.1_{\pm 1.1}$ (0.005) | $31.5_{\pm 0.9}$ (0.005) | $26.1_{\pm 2.8}$ (0.004) | $33.7_{\pm 2.4}$ (0.004) | $4.30_{\pm 1.5}$ (0.005) | $12.1_{\pm 4.4}$ (0.005) | $\mathbf{84.9}_{\pm 4.5}$ (0.005) | $\mathbf{83.7}_{\pm 4.0}$ (0.005) | $36.60_{\pm 2.5}$ (0.004) | $40.25_{\pm 2.9}$ (0.004) |
| Top-5 | $60.1_{\pm 1.4}$ (0.026) | $61.7_{\pm 1.8}$ (0.026) | $47.4_{\pm 3.6}$ (0.024) | $62.6_{\pm 3.3}$ (0.024) | $24.3_{\pm 3.0}$ (0.025) | $34.1_{\pm 2.6}$ (0.025) | $74.1_{\pm 6.6}$ (0.026) | $71.6_{\pm 5.4}$ (0.026) | $51.47_{\pm 3.7}$ (0.025) | $57.50_{\pm 3.3}$ (0.025) |
| Top-10 | $66.9_{\pm 0.6}$ (0.053) | $70.8_{\pm 2.3}$ (0.053) | $53.7_{\pm 3.7}$ (0.049) | $67.5_{\pm 2.3}$ (0.049) | $29.2_{\pm 5.1}$ (0.050) | $42.6_{\pm 3.2}$ (0.050) | $70.3_{\pm 0.9}$ (0.052) | $64.3_{\pm 7.1}$ (0.052) | $55.02_{\pm 2.6}$ (0.051) | $61.30_{\pm 3.7}$ (0.051) |
| Top-25 | $71.3_{\pm 2.7}$ (0.133) | $80.0_{\pm 1.2}$ (0.133) | $54.5_{\pm 4.6}$ (0.124) | $75.3_{\pm 1.9}$ (0.124) | $24.8_{\pm 2.2}$ (0.126) | $48.1_{\pm 2.5}$ (0.126) | $65.1_{\pm 12.0}$ (0.131) | $55.6_{\pm 7.8}$ (0.131) | $53.92_{\pm 5.4}$ (0.128) | $64.75_{\pm 3.4}$ (0.128) |
| Top-50 | $67.9_{\pm 4.1}$ (0.267) | $80.3_{\pm 0.5}$ (0.267) | $52.4_{\pm 3.9}$ (0.249) | $74.3_{\pm 2.0}$ (0.249) | $33.6_{\pm 3.6}$ (0.252) | $52.4_{\pm 4.7}$ (0.252) | $61.2_{\pm 11.8}$ (0.262) | $52.7_{\pm 7.4}$ (0.262) | $53.77_{\pm 5.9}$ (0.257) | $64.92_{\pm 3.7}$ (0.257) |
| LC | $56.2_{\pm 10.1}$ (1.000) | $88.0_{\pm 2.3}$ (1.000) | $45.1_{\pm 6.4}$ (1.000) | $76.9_{\pm 2.2}$ (1.000) | $23.8_{\pm 5.9}$ (1.000) | $56.6_{\pm 4.0}$ (1.000) | $49.5_{\pm 11.3}$ (1.000) | $58.4_{\pm 9.8}$ (1.000) | $43.65_{\pm 8.4}$ (1.000) | $69.97_{\pm 4.6}$ (1.000) |
| RAG | $71.6_{\pm 1.5}$ (0.026) | $75.2_{\pm 2.1}$ (0.026) | $49.8_{\pm 3.6}$ (0.024) | $65.8_{\pm 2.3}$ (0.024) | $26.4_{\pm 4.4}$ (0.025) | $42.6_{\pm 2.3}$ (0.025) | $70.9_{\pm 6.5}$ (0.026) | $74.1_{\pm 6.5}$ (0.026) | $54.67_{\pm 4.0}$ (0.025) | $64.42_{\pm 3.3}$ (0.025) |
| Self-Route | $65.8_{\pm 2.6}$ (0.187) | $80.0_{\pm 3.2}$ (0.282) | $51.3_{\pm 4.6}$ (0.181) | $59.7_{\pm 2.0}$ (0.249) | $25.3_{\pm 0.2}$ (0.381) | $52.4_{\pm 5.0}$ (0.625) | $56.4_{\pm 11.3}$ (0.888) | $57.5_{\pm 9.7}$ (0.943) | $49.70_{\pm 4.7}$ (0.409) | $62.40_{\pm 5.0}$ (0.524) |
| Adaptive-$k$ | $47.7_{\pm 4.7}$ (0.398) | $64.4_{\pm 2.3}$ (0.398) | $32.1_{\pm 6.0}$ (0.405) | $54.1_{\pm 1.4}$ (0.405) | $18.5_{\pm 3.0}$ (0.675) | $44.4_{\pm 1.8}$ (0.675) | $66.0_{\pm 8.3}$ (0.502) | $68.3_{\pm 6.8}$ (0.502) | $41.07_{\pm 5.6}$ (0.495) | $57.80_{\pm 3.1}$ (0.495) |
| BGM | $68.9_{\pm 1.6}$ (0.017) | $72.2_{\pm 3.0}$ (0.019) | $56.5_{\pm 3.1}$ (0.013) | $66.8_{\pm 0.8}$ (0.017) | $30.3_{\pm 4.8}$ (0.015) | $34.2_{\pm 2.2}$ (0.020) | $73.1_{\pm 6.1}$ (0.016) | $73.4_{\pm 6.0}$ (0.015) | $57.20_{\pm 3.9}$ (0.015) | $61.65_{\pm 3.0}$ (0.018) |
| RankZephyr | $56.1_{\pm 2.1}$ (0.026) | $62.7_{\pm 0.9}$ (0.026) | $54.5_{\pm 3.3}$ (0.024) | $71.4_{\pm 1.9}$ (0.024) | $20.9_{\pm 4.5}$ (0.025) | $28.0_{\pm 3.6}$ (0.025) | $73.3_{\pm 5.9}$ (0.026) | $75.8_{\pm 6.5}$ (0.026) | $51.20_{\pm 4.0}$ (0.025) | $59.48_{\pm 3.2}$ (0.025) |
| LDAR | $\mathbf{77.3}_{\pm 2.2}$ (0.209) | $\mathbf{90.5}_{\pm 0.8}$ (0.502) | $\mathbf{61.7}_{\pm 2.3}$ (0.272) | $\mathbf{82.7}_{\pm 0.8}$ (0.444) | $\mathbf{42.9}_{\pm 2.3}$ (0.312) | $\mathbf{65.8}_{\pm 1.4}$ (0.606) | $64.3_{\pm 10.5}$ (0.209) | $65.9_{\pm 6.3}$ (0.519) | $\mathbf{61.55}_{\pm 4.3}$ (0.250) | $\mathbf{76.22}_{\pm 2.3}$ (0.517) |

Hallucination task evaluates whether an LLM can appropriately refuse to answer when the provided context lacks the required information (which is always the case in this task). We observed that training models with feedback from the Hallucination task leads to undesirable strategies in practice (e.g., deliberately retrieving no passage or irrelevant passages to trigger refusal. See Appendix D.1 for more details). This behavior arises because the Hallucination task, by design, rewards avoidance rather than the constructive use of retrieved evidence, leading to degenerate strategies that do not reflect real-world retrieval requirements. To circumvent this issue, we treated the Hallucination task purely as an evaluation benchmark rather than a training objective. Consequently, for this task, we employed models trained on the Location task—the most widely adopted retrieval strategy in practice—and evaluated them in a zero-shot setting on the Hallucination task using the full dataset.

**HotpotQA** (Yang et al., 2018) and **Natural Questions** (NQ) (Kwiatkowski et al., 2019) are widely used open-domain QA benchmarks that are incorporated into the HELMET benchmark (Yen et al., 2024). Within HELMET, these datasets are adapted for long-context evaluation by extending input lengths to approximately 128K tokens through the inclusion of distractor passages, with all contexts drawn from Wikipedia. Among them, HotpotQA is distinguished as a multi-hop QA task, requiring the model to retrieve and integrate information from multiple passages to derive the correct answer.

## 5.2 BASELINES AND EXPERIMENTAL SETTINGS

**Baselines**    We compare LDAR with eight top-$k$ retrieval methods (Top-1, Top-5, Top-10, Top-25, Top-50, long-context, RAG (Lewis et al., 2020), RankZephyr (Pradeep et al., 2023)), one baseline designed to minimize the gap between a pretrained retriever and a pretrained LLM (BGM (Ke et al., 2024)), and two retrieval baselines that aim to balance the trade-off between RAG and long-context processing (Self-Route (Li et al., 2024b), Adaptive-$k$ (Taguchi et al., 2025)). For the top-$k$ retrieval baselines, we retrieved the top-$k$ passages according to similarity scores. Note that retrieving all passages (i.e., the full document context) corresponds to the long-context (LC) approach. For RAG, we applied a bge-reranker-large to reorder the retrieved top-5 passages. BGM trains a sequence-to-sequence model on textual information to identify the effective passage set among the top-5 candidates retrieved by similarity. Self-Route queries the LLM to decide between RAG and long-context processing based on the model's self-assessment of answerability. Adaptive-$k$ retrieves passages by identifying the largest gap in the sorted similarity score. The listwise reranker RankZephyr jointly processes the top-50 similarity-retrieved passages and produces a global ranking over them, from which it selects the final top-5 passages.

**Experimental Settings**    Following the evaluation metric used in LaRA (Li et al., 2025a) and HEL-MET (Yen et al., 2024), we employed GPT-4o to judge response correctness by providing it with the query, the ground-truth answer, and the model prediction. Throughout the experiments, we used bge-large-en-v1.5 as the embedding model. Note LDAR does not use reranker for reordering the retrieved passages, both to maintain high training efficiency and to demonstrate the effectiveness of our method in isolation. For open-source LLMs, we used Qwen-2.5-7B-Instruct, Qwen-3-4B-2507, Llama-3.1-8B-Instruct, Llama-3.2-3B-Instruct and Mistral-Nemo-Instruct-12B to evaluate the results. For closed-source LLMs, we used GPT-4o, GPT-4o-mini, Gemini-2.5-pro, Gemini-2.5-flash to evaluate the results. Across all LDAR experiments, we used the same hyperparameter configuration, as summarized in Appendix C.2.

## 5.3 MAIN RESULTS

Table 1 demonstrates the performance of LDAR compared to baseline methods. To ensure statistical significance, we report the average score across LLMs with standard errors, and provide separate averages for open-source and closed-source models as introduced in Section 5.2. LDAR generally achieves significantly higher performance compared to all other baselines, while using only about half the token usage of the long-context approach. It is worth noting that no penalty on token usage was imposed during the training of LDAR; the model was optimized solely for prediction accuracy (see Appendix D.5 for the cost-regularized variant of LDAR). These results suggest that a trade-off between information coverage and distraction does exist, and that LDAR is able to balance such trade-offs by leveraging the similarity distribution between the query and the passages. As LDAR typically retrieves more passages than RAG but fewer than the long-context approach, it achieves Hallucination scores that are generally higher than those of the long-context approach yet lower than those of RAG. This trend arises because retrieving a larger number of passages increases the likelihood of including misleading but seemingly relevant passages, thereby increasing the risk of hallucination. Nevertheless, LDAR consistently outperforms RAG across all other tasks, resulting in a significantly higher overall performance.

The average token usage ratio of LDAR relative to the long-context approach is: 0.47 (32K open-source), 0.63 (32K closed-source), 0.25 (128K open-source), 0.52 (128K closed-source). LDAR tends to use more tokens for closed-source models, which generally exhibit stronger long-context capability than open-source models. Notably, when the context length extends to 128K, the LDAR framework adapts by retrieving smaller portion of passages compared to the context length 32K setting, indicating that the risk of having distraction with the long-context approach increases with longer input contexts. These results indicate that the LDAR framework dynamically aligns its retrieval strategy to the long-context capability of the underlying LLM, adaptively balancing information coverage against potential distraction.

Breaking down by task, the average token usage ratio relative to the long-context approach is: 0.45 (Location), 0.43 (Reasoning), 0.50 (Comparison), 0.42 (Hallucination). Since the Comparison task requires integrating information from multiple regions of a whole context, LDAR is optimized to

retrieve a larger number of passages relative to other tasks. This highlights that the optimal retrieval strategy varies across tasks, and our framework effectively adapts its retrieval strategy accordingly.

## 5.4 Comparison to Baseline Methods

As demonstrated in Table 1, LDAR achieves significantly better performance compared to the Top-$k$ baselines, implying that LDAR executes a different retrieval strategy based on the similarity distribution between the query and passages (see Figure 3 (left) for visualizations). Heuristic baseline methods that try to balance the trade-off between RAG and long-context processing (Self-Route, Adaptive-$k$) fail to retrieve passages based on the long-context capability of LLMs, leading to worse performance. The reranker baselines (RAG, BGM, RankZephyr) also yield limited performance improvements, as they operate solely within the top-similarity region and only reorder those passages, effectively disregarding the LLM's long-context reasoning capability. Specifically, the learning method that selects the top-$k$ passages by similarity and subsequently identifying the optimal subset based on textual information through evaluation signals (BGM) yield only marginal gains in long-context settings. While increasing $k$ might seem like a straightforward solution, it substantially enlarges the combinatorial subset selection space, causing the model to converge to a suboptimal strategy. Accordingly, we report the best performance of BGM with $k = 5$, which is consistent with the original paper setting. In contrast, LDAR is explicitly designed for scalability by reducing the search space to a low-dimensional, smooth control space (Section 4.1), resulting in significantly better performance across overall settings.

## 5.5 Zero-shot Evaluation of LDAR

Table 2 evaluates whether retrieval strategies learned in the LaRA benchmark can generalize in a zero-shot manner to tasks in another benchmark (HELMET). To ensure alignment across multi-hop and single-hop tasks, LDAR trained on the Comparison task is evaluated zero-shot on long-context HotpotQA task, and LDAR trained on the Location task is evaluated zero-shot on long-context NQ task. Although the observed performance gap is smaller than in Table 1, LDAR still achieves better average performance compared to RAG or the long-context approach, while also attaining a lower token usage ratio relative to the long-context approach. These results indicate that LDAR's learned retrieval strategies can generalize to tasks in another benchmark in a zero-shot manner.

Table 2: Zero-shot performance results of LDAR on HotpotQA and NQ dataset. White cells denote LC, gray denotes RAG , and red denotes LDAR . Numbers in parentheses show token-usage ratio relative to LC. The best performing strategy for each task within each LLM is highlighted in bold.

| Method | HotpotQA | | | NQ | | |
|---|---|---|---|---|---|---|
| Llama-3.1-8B-Instruct | 52.0 (1.000) | 50.0 (0.019) | **59.0** (0.499) | 43.0 (1.000) | 40.0 (0.021) | **49.0** (0.213) |
| Llama-3.2-3B-Instruct | 54.0 (1.000) | 52.0 (0.019) | 54.0 (0.207) | 42.0 (1.000) | 37.0 (0.021) | **43.0** (0.146) |
| Qwen-2.5-7B-Instruct | 30.0 (1.000) | 63.0 (0.019) | **64.0** (0.305) | 25.0 (1.000) | 42.0 (0.021) | **54.0** (0.126) |
| Qwen-3-4B-Instruct | 56.0 (1.000) | 51.0 (0.019) | **62.0** (0.536) | 54.0 (1.000) | 41.0 (0.021) | 53.0 (0.486) |
| Mistral-Nemo-12B | 29.0 (1.000) | **63.0** (0.019) | 61.0 (0.061) | 23.0 (1.000) | 45.0 (0.021) | **47.0** (0.099) |
| **Open-source Average** | 44.2 (1.000) | 55.8 (0.019) | **60.0** (0.321) | 37.4 (1.000) | 41.0 (0.021) | **49.2** (0.214) |
| GPT-4o | 81.0 (1.000) | 65.0 (0.019) | **84.0** (0.579) | 61.0 (1.000) | 54.0 (0.021) | 60.0 (0.738) |
| GPT-4o-mini | 64.0 (1.000) | 65.0 (0.019) | **76.0** (0.629) | 52.0 (1.000) | 52.0 (0.021) | **59.0** (0.374) |
| Gemini-2.5-pro | **85.0** (1.000) | 55.0 (0.019) | 84.0 (0.638) | 62.0 (1.000) | 37.0 (0.021) | **65.0** (0.518) |
| Gemini-2.5-flash | 82.0 (1.000) | 57.0 (0.019) | **83.0** (0.953) | 54.0 (1.000) | 37.0 (0.021) | **60.0** (0.564) |
| **Closed-source Average** | 78.0 (1.000) | 60.5 (0.019) | **81.8** (0.699) | 59.0 (1.000) | 45.0 (0.021) | **61.0** (0.457) |

## 5.6 Cost Efficiency Analysis of LDAR

While LDAR brings performance improvements, it introduces additional cost, both from its training procedure and from the extra inference-time incurred by the forward pass of the learned retriever $\pi_\theta$. To quantify this overhead, we measured the end-to-end inference time per example in Location task with all baseline methods. Specifically, the inference time includes the time required to (1) compute the query and passage embeddings, (2) run the bridge model (e.g., reranker or LDAR retriever), and (3) process the selected passages through the prediction LLM to generate the final answer.

Table 3 summarizes the average per-example inference time with corresponding performance across all baselines using both open-source LLMs and closed-source LLMs. With open-source models, LC exhibits the highest latency as it forwards all retrieved passages to the underlying LLM. Methods that rely on text-based rerankers (RAG, RankZephyr, BGM) also incur substantial overhead due to heavy cross-encoder computation. In contrast, LDAR employs a lightweight, text-free adaptive re-

Table 3: Comparison of retrieval strategies on LaRA Location task. Each cell reports the average per-example inference time or performance with standard error, computed using open-source and closed-source LLMs. Numbers in parentheses denote standard error for inference time and the token-usage ratio for performance metrics, respectively.

| Metric | Top-1 | Top-5 | Top-10 | Top-25 | Top-50 | LC | RAG | Self-Route | Adaptive-$k$ | BGM | Rank Zephyr | LDAR |
|---|---|---|---|---|---|---|---|---|---|---|---|---|
| Time (Open-source) | 1.3 (0.55) | 1.5 (0.73) | 1.7 (0.91) | 2.5 (0.99) | 4.0 (1.31) | 15.4 (5.17) | 18.4 (0.86) | 4.5 (5.93) | 8.6 (11.70) | 13.3 (1.14) | 10.2 (1.07) | 3.9 (1.61) |
| Time (Closed-source) | 3.9 (2.29) | 4.7 (1.91) | 5.2 (2.36) | 5.8 (2.52) | 5.9 (2.20) | 8.2 (2.67) | 22.6 (1.75) | 9.0 (4.09) | 6.5 (4.70) | 17.8 (5.94) | 13.6 (2.56) | 8.0 (2.69) |
| Score (Open-source) | 31.1 (0.01) | 60.1 (0.03) | 66.9 (0.05) | 71.3 (0.13) | 67.9 (0.27) | 56.2 (1.00) | 71.6 (0.03) | 65.8 (0.19) | 47.7 (0.40) | 68.9 (0.02) | 56.1 (0.03) | **77.3** (0.21) |
| Score (Closed-source) | 31.5 (0.01) | 61.7 (0.03) | 70.8 (0.05) | 80.0 (0.13) | 80.3 (0.27) | 88.0 (1.00) | 75.2 (0.03) | 80.0 (0.28) | 64.4 (0.40) | 72.2 (0.02) | 62.7 (0.03) | **90.5** (0.50) |

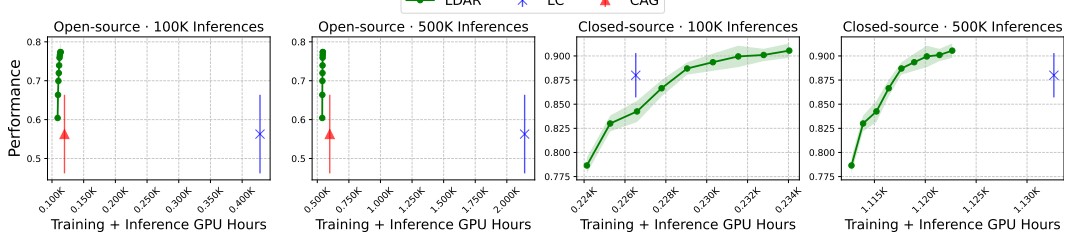

Figure 5: Average performance plotted against total cumulative computational cost (GPU hours) for both training and inference. The left two panels report the total cost when applying LDAR at different training epochs under 100K and 500K inference calls using **open-source LLMs**; the right two panels show the corresponding results for **closed-source LLMs**.

trieval mechanism, selecting passages to minimize potential interference from distracting passages. As a result, LDAR achieves the fastest inference time among LC and all reranker baselines, while simultaneously achieving the best overall performance. In the closed-source LLMs, the underlying LLM is highly optimized for fast inference, making total inference time less sensitive to input length. Even under these conditions, LDAR remains faster than LC and all reranker baselines while also achieving better performance.

In addition, we include an analysis illustrating how the cost benefits accumulate over time when deploying the learned lightweight adaptive retriever $\pi_\theta$ in Figure 5. For each LDAR training epoch, we compute (1) the cumulative training cost up to that epoch and (2) the inference cost when deploying $\pi_\theta$ learned until that epoch with a certain number of inference calls (100K and 500K inference calls). We include CAG (Chan et al., 2025), a cache-augmented generation method that accelerates inference by preloading documents and reusing a precomputed KV-cache, as an additional baseline to compare inference-time efficiency with LDAR.

Since LDAR has lower inference-time overhead compared to LC, its cost advantage becomes increasingly pronounced as the number of deployments grows. As LDAR trains, it learns to identify the most effective passage set that minimizes distraction. This not only improves performance but also reduces token usage during both training and inference. As shown in the right panels of Figure 5, LDAR progressively selects fewer tokens over training epochs, thereby reducing the per-epoch training cost. These results demonstrate that LDAR ensures a favorable cost–benefit trade-off in practice.

## 6 CONCLUSION

In this paper, we demonstrated retrieving passages to minimize distraction remains a challenging problem, as the optimal strategy depends on both the capacity of the target LLM and the interactions among retrieved passages. Although a high-capacity LLM could exhaustively align all passages with the query to minimize distraction, the cost grows prohibitively with passage count, making this approach impractical. To this end, we present LDAR, an adaptive retriever that selects passages in accordance with LLM's long-context capability to minimize potential interference from distracting passages, relying solely on the similarity distribution. Experiments across diverse LLM architectures and knowledge-intensive benchmarks demonstrate that LDAR achieves significantly better performance compared to baselines with lower token usage compared to the long-context approach.

## 7 ACKNOWLEDGEMENTS

This work was partly supported by Institute of Information & Communications Technology Planning & Evaluation (IITP) grant funded by the Korea government (MSIT) (No. RS-2022-II220311, Development of Goal-Oriented Reinforcement Learning Techniques for Contact-Rich Robotic Manipulation of Everyday Objects (31%), No. RS-2024-00457882, AI Research Hub Project, No. RS-2019-II190079, Artificial Intelligence Graduate School Program (Korea University), the IITP (Institute of Information & Communications Technology Planning & Evaluation)-ITRC (Information Technology Research Center) grant funded by the Korea government (Ministry of Science and ICT) (IITP-2025-RS-2024-00436857) (31%), the NRF (RS-2024-00451162) funded by the Ministry of Science and ICT, Korea, BK21 Four project of the National Research Foundation of Korea, the National Research Foundation of Korea (NRF) grant funded by the Korea government (MSIT) (RS-2025-00560367), the IITP under the Artificial Intelligence Star Fellowship support program to nurture the best talents (IITP-2025-RS-2025-02304828) (32%) grant funded by the Korea government (MSIT), and KOREA HYDRO & NUCLEAR POWER CO., LTD (No. 2024-Tech-09).

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

## A  THE USE OF LLM IN PAPER WRITING

We utilized large language models to improve the clarity and phrasing of the text.

## B  LIMITATIONS AND FUTURE WORK

While LDAR was trained using task-specific signal, retrieval scenarios in practice are often diverse and may not align neatly with a single task formulation. As our results in Section 5.3 indicate that the optimal retrieval strategy varies across tasks, a promising direction for future research would be to develop a meta-classifier capable of identifying the underlying retrieval task and employing a mixture-of-experts framework (Shazeer et al., 2017), where task-specialized retrieval strategies are adaptively combined.

Moreover, LDAR neither explicitly models the ordering of retrieved passages nor employs a reranker to reorder them, which may affect downstream performance. While rerankers were deliberately excluded to maintain high training efficiency, an important direction is to explore learning-based retrieval strategies that jointly optimize both passage selection and ordering in long-context settings.

## C  EXPERIMENTAL DETAILS

### C.1  IMPLEMENTATION DETAILS OF THE ADAPTIVE RETRIEVAL PROCESS

We employ periodic embedding layer (Gorishniy et al., 2022) for encoding numeric features, which projects batch of raw scalar inputs $s \in \mathbb{R}^{B \times N \times 1}$ to the embedding dimension $\mathbb{R}^{B \times N \times d}$, followed by layer normalization. The embedded tokens are then processed by a self-attention Transformer model and outputs $h \in \mathbb{R}^{B \times N \times D}$. A learned linear token scorer projects each Transformer output to a scalar, which is normalized with a softmax to obtain attention weights $w \in \mathbb{R}^{B \times N}$. We then form a global summary by attention pooling over the token dimension $N$: $z_{b,d} = \sum_{n=1}^{N} w_{b,n} h_{b,n,d}$, and pass $z \in \mathbb{R}^{B \times D}$ through a small MLP head to obtain g. From g, four linear heads produce the parameters of two Beta distributions: $(\alpha_L, \beta_L)$ for the lower quantile and $(\alpha_\Delta, \beta_\Delta)$ for the band width $q_\Delta$ to ensure that lower quantile $q_L$ to be smaller than upper quantile $q_U$. In addition, we apply softplus to ensure all parameters of Beta distributions are strictly positive.

We train LDAR on 4 NVIDIA RTX 3090 GPUs for 32K context length settings and 1 NVIDIA RTX PRO 6000 GPU for 128K context length settings.

### C.2  HYPERPARAMETER SETTINGS

Our hyperparameter settings are summarized in Table 4. The same configuration is used consistently across all experiments.

Table 4: Hyperparameter settings used for LDAR

| Hyperparameter | Setting |
|---|---:|
| Batch Size | 32 |
| Embedding Dimension | 256 |
| Transformer Hidden Dimension | 256 |
| Baseline EMA coefficient | 0.5 |
| # Transformer Layer | 2 |
| # Transformer Head | 4 |
| Optimizer | Adam($\beta = [0.9, 0.999], \epsilon = $ 1e-8) |
| learning rate $\gamma$ | 3e-4 |

### C.3  IMPLEMENTATION DETAILS OF TEXT CHUNKING PROCEDURE

We followed the standard LaRA benchmark chunking procedure, forming 600-token passages with 100-token overlap. As the size of each passage is equal in length, we can compute the number

of passages directly from the information provided in Table 1. This results in approximately 64 passages for the 32k context-length setting and 256 passages for the 128k setting, which are used consistently across all methods. Since the reported token-usage ratio corresponds exactly to the fraction of passages retrieved, the number of passages retrieved by each baseline can be computed directly from the table.

### C.4 IMPLEMENTATION DETAILS OF BASELINE METHODS

**Self-Route** Li et al. (2024b) leverages the LLM itself to decide whether to use the RAG or long-context approach. The method consists of two simple steps: (1) the query and Top-$k$ retrieved passages are provided to the LLM, which is prompted to predict whether the query can be answered with the given passages; (2) if the LLM predicts the query is answerable, the LLM generates the answer directly. Otherwise, the full passage pool is passed to the LLM to produce the final prediction using the long-context approach. To implement this routing process, we adapt the prompt used in the LaRA benchmark (Li et al., 2025a), following the design of Self-Route (see Appendix C in Li et al., 2024b).

> Here are some chunks retrieved from some {datatype}. Read these chunks to answer a question. Be concise. If the question cannot be answered based on the information in the article, write "unanswerable". {context} Question: {question} Only give me the answer and do not output any other words. If the question cannot be answered based on the information in the article, write "unanswerable". Answer:

Table 5: Prompt we used for Self-Route baseline.

**Adaptive-$k$** Taguchi et al. (2025) retrieves passages by locating the largest gap in the sorted similarity scores. Following the pseudo-code described in the original paper, we compute the difference between consecutive similarity scores in sorted order, identify the index corresponding to the maximum gap, and retrieve all passages preceding this index.

**BGM** Ke et al. (2024) proposed a learning method that first selects the top-$k$ passages based on similarity scores and subsequently identifies the optimal subset using textual information guided by evaluation learning signals. Since the official implementation is not publicly available, we implemented BGM by following the procedure described in the paper. Since the paper does not specify the number of silver passages required for the supervised learning stage, we constructed 50 silver passages per task and used the evaluation signal to further fine-tune the sequence-to-sequence model.

**CAG** Chan et al. (2025) operates by first computing a KV-cache over a combined set of documents $D = \{d_1, d_2, \ldots\}$ that corresponds to a set of queries $\{q_1, q_2, \ldots\}$. Once this large unified KV-cache is constructed, the model can process each instance efficiently by reusing the cached representations. In their experiments on HotPotQA, the largest-scale setting aggregates 64 documents (approximately 85k tokens) to build this cache. However, in long-context benchmarks such as LaRA, a single document associated with a single query already reaches the context-length limit of LLMs (which is 128k tokens in our experiments). Therefore, computing a KV-cache over multiple documents is infeasible under long-context setting, and **CAG effectively exhibits the same computational complexity as standard LC**. That said, the LaRA benchmark has a particular characteristic: each document is paired with multiple queries (e.g., 10 queries per document). This enables CAG to gain some benefit by caching individual documents and reusing their cached representations across queries that reference the same document. Accordingly, in our experiments, we implemented CAG by caching each document separately and applying the cache to all associated queries. We report results based on this fair and practically feasible adaptation of CAG for long-context evaluation. With such experimental settings, CAG achieved performance comparable to LC, but yields a 2.5× reduction in latency (15.41 seconds for LC vs. 4.31 seconds for CAG per inference on average).

# D    ADDITIONAL EXPERIMENT RESULTS

## D.1    TRAINING LDAR ON HALLUCINATION TASK

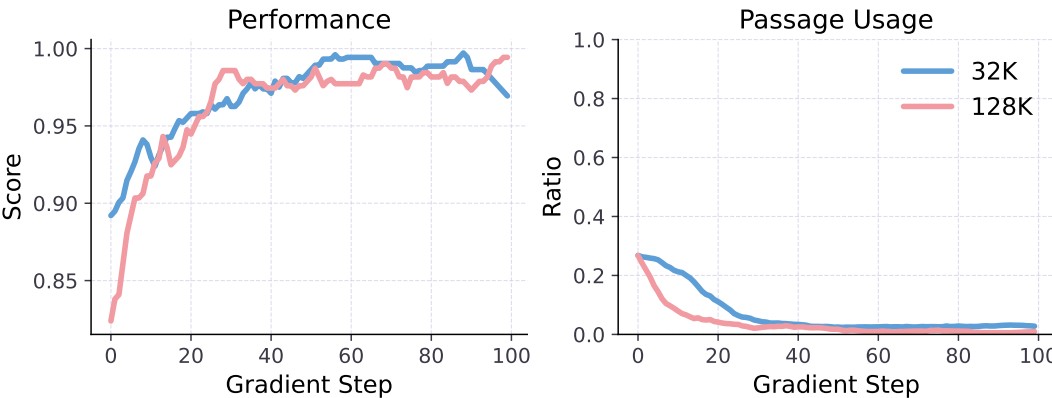

Figure 6: Performance and passage-usage ratio of LDAR trained with the Hallucination task signal. Results are shown for Llama-3.1-8B under 32K and 128K context-length settings.

We visualize the training curve of LDAR when trained with the evaluation signal from the Hallucination task in the LaRA benchmark. The Hallucination task measures whether an LLM can correctly refuse to answer when the provided context lacks the required information (which is always the case in this task). As shown in Figure 6, LDAR quickly learns to retrieve almost no passages, thereby forcing the LLM to refuse answering. This behavior results in degenerate strategies that do not reflect realistic retrieval requirements. Consequently, we treat the Hallucination task purely as an evaluation benchmark rather than a training objective. We employed models trained on the Location task–the most widely adopted retrieval strategy in practice–to evaluate and report the performance of LDAR in Hallucination task at Table 1.

## D.2 VISUALIZATION OF RETRIEVAL STRATEGIES LEARNED BY LDAR

Figure 7 shows additional visualizations of retrieval strategies learned by LDAR. LDAR adaptively determines both the lower and upper bounds of the similarity distribution band used for passage retrieval, thereby maximizing LLM prediction accuracy by balancing information coverage against the risk of distraction.

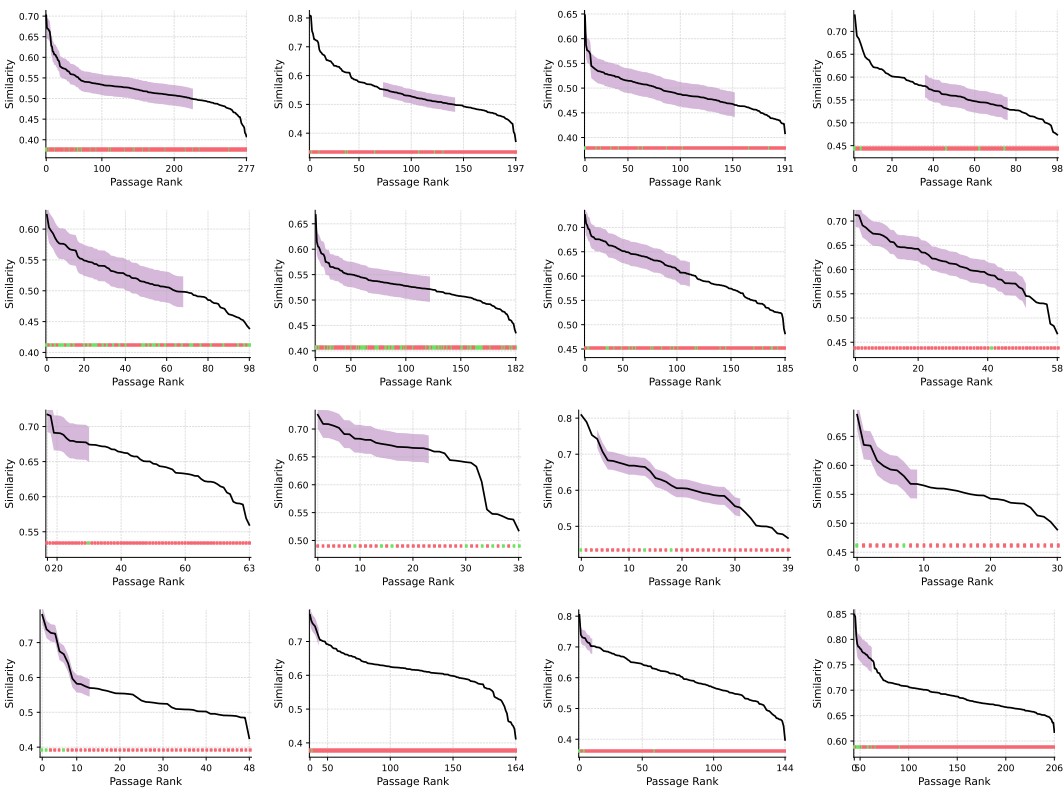

Figure 7: Visualization of passages retrieved by LDAR based on the similarity distribution, with retrieved passages marked in green if they contain the correct answer and in red otherwise.

**Case study of LDAR's Retrieval Strategy**    Figure 8 shows a case-study analyzing LDAR's learned retrieval strategies. When the similarity distribution shows a clear high-similarity region, LDAR tends to focus narrowly on that region. In these cases, the useful passage tends to be located within the top rank passages as illustrated in Figure 8 (top).

Conversely, when the overall similarity distribution is low in value and relatively flat in shape, LDAR expands its retrieval interval to ensure broader coverage, even at the risk of including more distracting passages. In these cases, the useful passage is often not located within the top rank passages as illustrated in Figure 8 (bottom), making a wider retrieval band necessary.

Although LDAR does not receive any textual information, this case study demonstrates that the shape of the similarity distribution is indicative of where useful passages tend to appear. LDAR learns these patterns and adapts its retrieval strategy accordingly.

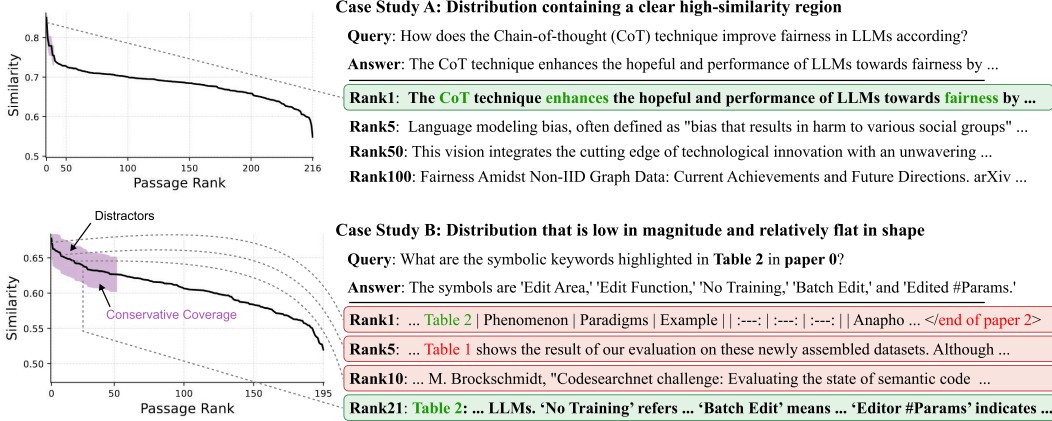

Figure 8: A case study illustrating LDAR's learned retrieval behavior. When the similarity distribution contains a clear high-similarity region, LDAR tends to focus narrowly on that region. When the overall similarity distribution is low in value and relatively flat in shape, LDAR expands its retrieval interval to ensure broader coverage, even at the risk of including more distracting passages.

## D.3 ANALYSIS OF LDAR'S TRAINING / INFERENCE EFFICIENCY

LDAR introduces a small additional computation during inference because it performs a forward pass of the learned LDAR retrieval strategy $\pi_\theta$. To quantify this overhead, we measured the end-to-end inference time per example in Location task. Specifically, the inference time includes the time required to (1) compute the query and passage embeddings, (2) run the bridge model (e.g., reranker or LDAR retriever), and (3) process the selected passages through the prediction LLM to generate the final answer. Note as baseline methods distribute their computation differently across these three stages, their total inference cost varies accordingly.

Table 6 summarizes the average per-example inference time across all methods using both open-source LLMs and closed-source LLMs. With open-source models, LC exhibits the highest latency because it forwards all retrieved passages to the prediction LLM. Methods such as RAG, RankZephyr, and BGM (which rely on text-based rerankers) also incur substantial overhead due to heavy cross-encoder computation. In contrast, LDAR employs a lightweight, text-free adaptive retrieval mechanism, selecting passages to minimize potential interference from distracting passages. As a result, LDAR achieves the fastest inference time among LC and all reranker baselines, while simultaneously achieving the best overall performance.

In the closed-source setting, the prediction LLM is highly optimized for fast inference, making total inference time less sensitive to input length. Even under these conditions, LDAR remains faster than LC and all reranker baselines while also achieving better performance. These results demonstrate that LDAR provides distraction-aware and long-context-capability-aware retrieval in a computationally efficient manner at inference time.

Table 6: Comparison of retrieval strategies on LaRA Location task. Each cell reports the average per-example inference time or performance with standard error, computed using open-source and closed-source LLMs. Numbers in parentheses denote standard error for inference time and the token-usage ratio for performance metrics, respectively.

| Metric | Top-1 | Top-5 | Top-10 | Top-25 | Top-50 | LC | RAG | Self-Route | Adaptive-$k$ | BGM | Rank Zephyr | LDAR |
|---|---|---|---|---|---|---|---|---|---|---|---|---|
| Time (Open-source) | 1.3 (0.55) | 1.5 (0.73) | 1.7 (0.91) | 2.5 (0.99) | 4.0 (1.31) | 15.4 (5.17) | 18.4 (0.86) | 4.5 (5.93) | 8.6 (11.70) | 13.3 (1.14) | 10.2 (1.07) | 3.9 (1.61) |
| Time (Closed-source) | 3.9 (2.29) | 4.7 (1.91) | 5.2 (2.36) | 5.8 (2.52) | 5.9 (2.20) | 8.2 (2.67) | 22.6 (1.75) | 9.0 (4.09) | 6.5 (4.70) | 17.8 (5.94) | 13.6 (2.56) | 8.0 (2.69) |
| Score (Open-source) | 31.1 (0.01) | 60.1 (0.03) | 66.9 (0.05) | 71.3 (0.13) | 67.9 (0.27) | 56.2 (0.19) | 71.6 (0.03) | 65.8 (1.00) | 47.7 (0.40) | 68.9 (0.02) | 56.1 (0.03) | **77.3** (0.21) |
| Score (Closed-source) | 31.5 (0.01) | 61.7 (0.03) | 70.8 (0.05) | 80.0 (0.13) | 80.3 (0.27) | 88.0 (1.00) | 75.2 (0.03) | 80.0 (0.28) | 64.4 (0.40) | 72.2 (0.02) | 62.7 (0.03) | **90.5** (0.50) |

Based on the analysis of LDAR's inference-time efficiency, we now illustrate how the cost benefits accumulate over time when deploying the learned LDAR strategy. For each LDAR training epoch, we compute (1) the cumulative training cost up to that epoch and (2) the inference cost when applying LDAR strategy learned until that epoch with a certain number of inference calls (10K, 100K, and 500K inference calls). We then plot the cumulative cost on the x-axis and the resulting performance on the y-axis. Figure 9 (top, bottom) shows the results for using open-source and closed-source LLMs are prediction models, respectively. Because LDAR has lower inference-time overhead compared to LC, its cost advantage becomes increasingly pronounced as the number of deployments grows.

Importantly, as LDAR trains, it learns to identify the most effective passage set that minimizes distraction. This not only improves performance but also reduces token usage during both training and inference. As shown in the center plot of Figure 9 (bottom), LDAR progressively selects fewer tokens over training epochs, thereby lowering the per-epoch training cost as well. CAG (Chan et al., 2025) is a cache-augmented generation method that speeds up inference by preloading documents and reusing a precomputed KV-cache. We included CAG as an additional baseline to compare inference-time efficiency with LDAR.

Additionally, to provide a fair comparison in terms of API usage, we converted all training and inference costs into USD and visualized the results in Figure 10. GPU hours are converted into USD based on the pricing of the cloud computing services we used, and API costs are computed according to the official pricing of OpenAI and Google. Under this USD-based metric, LDAR demonstrates greater cost efficiency than LC, particularly when closed-source LLMs are used as the prediction model, as the cost of processing long-context inputs is extremely high.

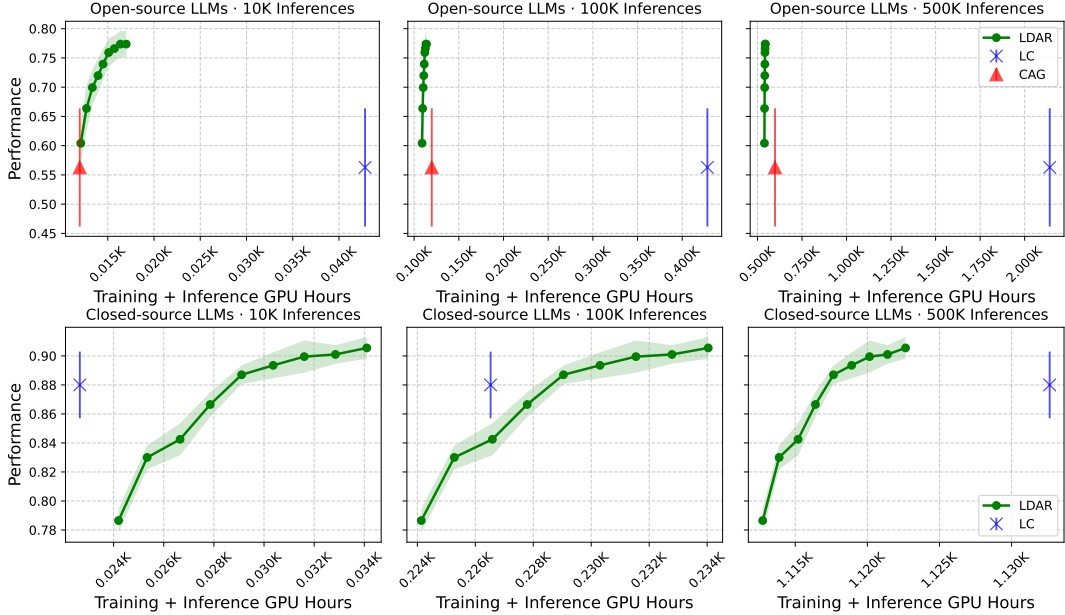

Figure 9: Average performance plotted against the total cumulative computational cost (GPU hours) for both training and inference. Each figure shows total cost incurred when using the LDAR retriever at different training epochs and running certain number of inference calls (10K, 100K, and 500K). The top row presents results using open-source LLMs, while the bottom row presents results using closed-source LLMs.

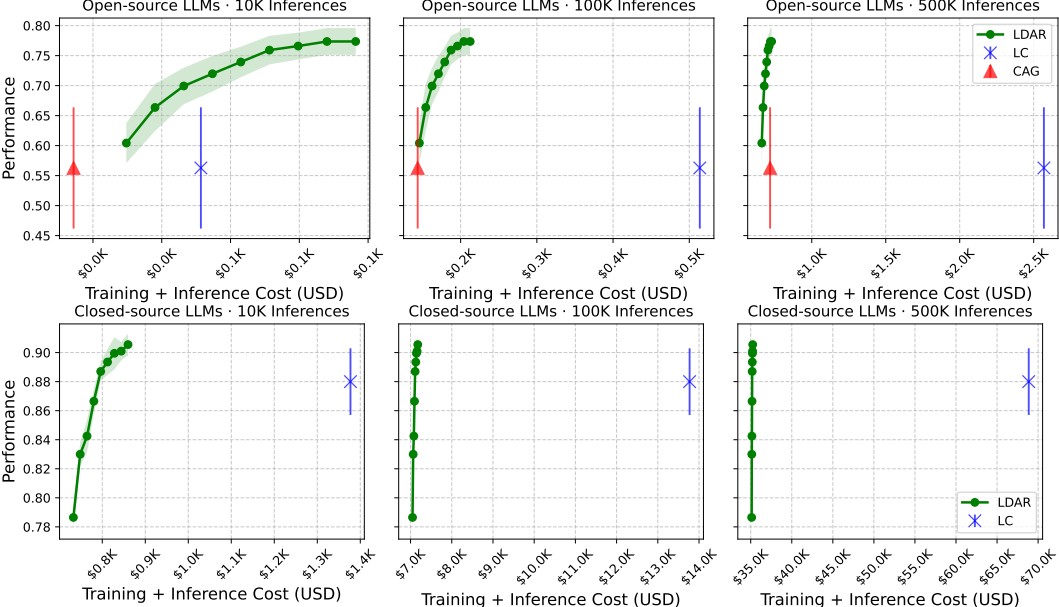

Figure 10: Average performance plotted against the total cumulative computational cost (USD) for both training and inference. We converted GPU hours into USD based on the pricng of cloud computing service we used, and API costs are computed according to the official pricing of OpenAI and Google. Each figure shows total cost incurred when using the LDAR retriever at different training epochs and running certain number of inference calls (10K, 100K, and 500K). The top row presents results using open-source LLMs, while the bottom row presents results using closed-source LLMs.

### D.4 CROSS-EMBEDDER GENERALIZATION

For an LDAR strategy trained with one embedding model to generalize to another, we found that a certain degree of scale alignment and rank alignment between the two embedders is necessary.

To illustrate this, Figure 11 (left) compares the mean cosine-similarity values produced by bge-large-en-v1.5 (Xiao et al., 2024) and gte-large-en-v1.5 (Zhang et al., 2024b) over 100 randomly sampled query–passage pairs from the Location task. Figure 11 (right) further shows how passages ranked by the bge-large-en-v1.5 map to their corresponding ranks in the gte-large-en-v1.5. These plots demonstrate that the two embedding models exhibit some amount of scale alignment (their similarity ranges are comparable) and rank alignment (higher-ranked passages in one model tend to remain relatively high-ranked in the other. Correlation: 0.7715). Because of this alignment, LDAR strategy trained with bge-large-en-v1.5 can generalize to gte-large-en-v1.5 to a reasonable extent, and vice versa, as shown in Table 7.

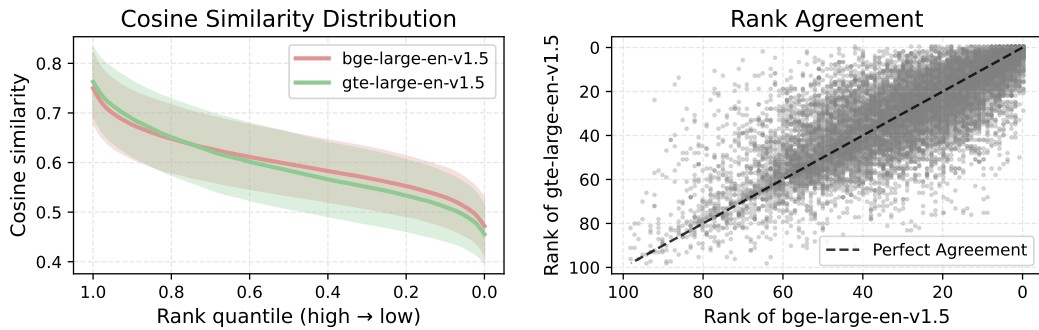

Figure 11: Comparison of cosine-similarity and rank agreement between bge-large-en-v1.5 and gte-large-en-v1.5 on the LaRA Location task. (Left) Average sorted cosine-similarity curve with standard deviation across all examples. (Right) Passage ranks under bge-large-en-v1.5 plotted against their ranks under gte-large-en-v1.5; the dashed line indicates perfect rank agreement between the two embedding models.

Table 7: Performance comparison across Location, Reasoning, Comparison, and Hallucination tasks.

(a) Location

| Embedding Model | LC | RAG | LDAR |
|---|---|---|---|
| bge-large-en-v1.5 | 69.3 (1.0) | 72.4 (0.03) | **77.6** (0.22) |
| gte-large-en-v1.5 | 69.3 (1.0) | 70.4 (0.02) | **74.4** (0.44) |
| bge → gte-large-en-v1.5 | 69.3 (1.0) | 70.4 (0.02) | **74.4** (0.30) |
| gte → bge-large-en-v1.5 | 69.3 (1.0) | 72.4 (0.03) | **75.5** (0.34) |

(b) Reasoning

| Embedding Model | LC | RAG | LDAR |
|---|---|---|---|
| bge-large-en-v1.5 | 50.6 (1.0) | 45.4 (0.02) | **59.6** (0.36) |
| gte-large-en-v1.5 | 50.6 (1.0) | 51.9 (0.02) | **57.1** (0.31) |
| bge → gte-large-en-v1.5 | 50.6 (1.0) | 51.9 (0.02) | **58.4** (0.25) |
| gte → bge-large-en-v1.5 | 50.6 (1.0) | 45.4 (0.02) | **54.5** (0.39) |

(c) Comparison

| Embedding Model | LC | RAG | LDAR |
|---|---|---|---|
| bge-large-en-v1.5 | 34.1 (1.0) | 19.5 (0.03) | **41.5** (0.50) |
| gte-large-en-v1.5 | 34.1 (1.0) | 24.3 (0.02) | **39.0** (0.54) |
| bge → gte-large-en-v1.5 | **34.1** (1.0) | 24.3 (0.02) | 26.8 (0.62) |
| gte → bge-large-en-v1.5 | **34.1** (1.0) | 19.5 (0.03) | 24.9 (0.47) |

(d) Hallucination

| Embedding Model | LC | RAG | LDAR |
|---|---|---|---|
| bge-large-en-v1.5 | 58.4 (1.0) | **78.3** (0.03) | 73.3 (0.22) |
| gte-large-en-v1.5 | 58.4 (1.0) | **77.9** (0.02) | 70.1 (0.44) |
| bge → gte-large-en-v1.5 | 58.4 (1.0) | **77.9** (0.02) | 71.4 (0.31) |
| gte → bge-large-en-v1.5 | 58.4 (1.0) | **78.3** (0.03) | 68.8 (0.36) |

(e) Average

| Embedding Model | LC | RAG | LDAR |
|---|---|---|---|
| bge-large-en-v1.5 | 53.1 (1.0) | 53.9 (0.03) | **63.0** (0.32) |
| gte-large-en-v1.5 | 53.1 (1.0) | 56.1 (0.02) | **60.1** (0.43) |
| bge → gte-large-en-v1.5 | 53.1 (1.0) | 56.1 (0.02) | **57.7** (0.37) |
| gte → bge-large-en-v1.5 | 53.1 (1.0) | 53.9 (0.03) | **55.9** (0.39) |

### D.5 TRAINING LDAR WITH COST-REGULARIZED OBJECTIVE

Figure 12 shows the optimization of LDAR if we included a penalty term proportional to the number of retrieved tokens during training. Introducing such a cost term caused the optimization to collapse into a local optimum, where the LDAR strategy retrieved too few passages and suffered a drop in accuracy.

Including a cost term would have been necessary if performance increases monotonically with the number of retrieved passages. However, both prior work (e.g., the inverted-U findings in (Jin et al., 2024; Leng et al., 2024)) and our own experiments show that performance peaks at an intermediate retrieval size and then decreases, forming an inverted-U relationship due to distraction from additional passages. Because the objective is not a linear trade-off but rather identifying the peak of this inverted-U, we found penalizing token usage during training often pushes the model away from the optimal region.

Therefore, instead of imposing an explicit cost term, we focused on letting LDAR learn to retrieve passages near the performance peak of the inverted-U. Notably, even without a cost term, LDAR naturally avoids the high-distraction region and achieves both higher accuracy and lower token usage compared to the long-context baseline (Table 1).

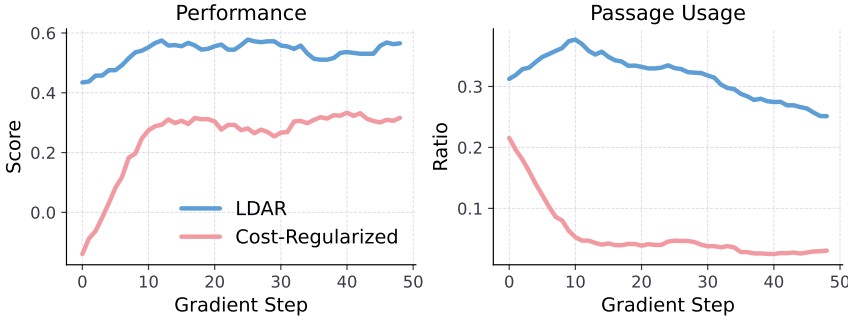

Figure 12: Performance and passage-usage ratio of LDAR with and without the cost-regularization term, evaluated on the LaRA Reasoning task using Qwen-2.5-7B.

### D.6 VISUALIZATION OF LDAR'S LEARNED RETRIEVAL STRATEGY IN CLUSTERS

To better understand LDAR's retrieval behavior, we performed a clustering analysis over LDAR's retrieval on both an open-source model (Qwen-2.5-7B) and a closed-source model (Gemini-2.5-pro) (Figure 13 and 14). We collected 100 similarity distributions and clustered them based on LDAR's retrieval band value, resulting in two primary clusters (Cluster 0 and Cluster 1). The figure summarizes the mean similarity distribution for each cluster along with LDAR's average retrieval band $(q_L, q_U)$.

The visualization demonstrates that when the similarity distribution contains a relatively high-similarity region, LDAR concentrates its retrieval on that narrow interval (Cluster 0). Conversely, when similarity values are relatively low, LDAR expands the retrieval range to increase information coverage (Cluster 1). To specifically examine cases where LDAR avoids the top interval, we additionally isolated all instances where the learned upper quantile satisfies $q_U < 1.0$. These cases typically fall within Cluster 1, but we regrouped them separately to analyze this behavior in finer detail. As illustrated in the third plot, these distributions exhibit the lowest overall similarity values.

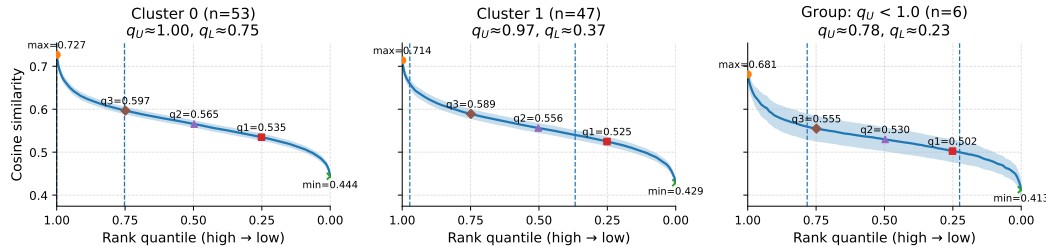

Figure 13: Visualization of clustering analysis over LDAR's retrieval behavior on **Qwen-2.5-7B** in **Location** task. We collected 100 similarity distributions and clustered them based on LDAR's retrieval band value, resulting in two primary clusters (Cluster 0 and Cluster 1). We additionally isolated all instances where the learned upper quantile satisfies $q_U < 1.0$. The figure shows the mean similarity distribution for each cluster along with LDAR's average retrieval band $(q_L, q_U)$. $n$ indicates the number of samples in each cluster and group.

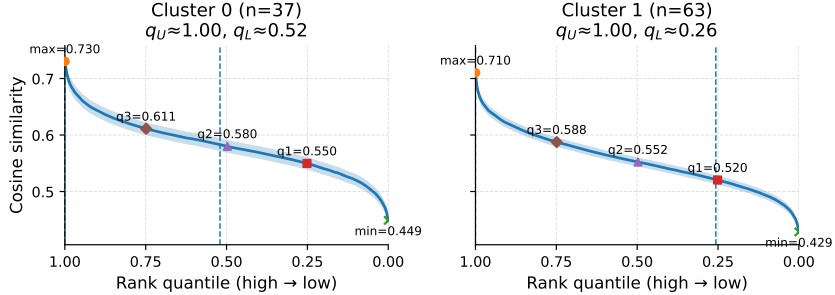

Figure 14: Visualization of clustering analysis over LDAR's retrieval behavior on **Gemini-2.5-pro** in **Location** task. We collected 100 similarity distributions and clustered them based on LDAR's retrieval band value, resulting in two primary clusters (Cluster 0 and Cluster 1). In this case, there is no instance where the learned upper quantile satisfies $q_U < 1.0$. The figure shows the mean similarity distribution for each cluster along with LDAR's average retrieval band $(q_L, q_U)$. $n$ indicates the number of samples in each cluster and group.

To better understand why LDAR band shifts downward in such cases, we performed a fixed-width sliding-window experiment (Figure 18). For each cluster, we sweep windows of size $w \in \{20, 50, 100\}$ across the similarity-ranked passages and measured accuracy using each window as the retrieved set. The sliding-window experiment demonstrates that when the similarity distribution contains a relatively high-similarity region (Cluster 0), narrower interval $(w = 20, 50)$ shows a better performance peak than wider interval $(w = 100)$. Conversely, when similarity values are relatively low (Cluster 1), wider interval $(w = 100)$ shows a better performance peak. This behavior

aligns with LDAR's learned retrieval bands. Interestingly, for Cluster 1 we observe a secondary performance peak when a wide window ($w = 100$) covers the mid-quantile region. This suggests that, in low-similarity scenarios, informative passages are often dispersed across the middle ranks rather than concentrated at the very top. We believe such occasional peaks in the mid-quantile region made LDAR to occasionally shifts its retrieval band downward to select mid-similarity passages when the overall similarity scale is low. The sliding-window experiment for the group $q_U < 1.0$ (Figure 18 (bottom)) further shows that the highest-ranked passage can sometimes act as a distractor when the overall similarity scale is low.

By comparing LDAR's behavior across open-source and closed-source LLMs (Figure 13 and 14), LDAR generally uses wider quantile bands when interacting with closed-source models, reflecting the fact that these models possess stronger long-context processing capabilities. Additionally, the frequency of non–top-interval retrieval (i.e., cases where $q_U < 1.0$) decreases notably for closed-source models. This indicates that closed-source LLMs are inherently more resilient to noisy or misleading passages, reducing the need for LDAR to shift its retrieval band downward to avoid distraction. Together, these observations highlight that LDAR internalizes and adapts to the long-context characteristics of the underlying LLM, producing retrieval behaviors that are both model-aware and capability-aligned.

## D.7 Visualization of LDAR's Zero-Shot Retrieval on HELMET and Ada-LEval

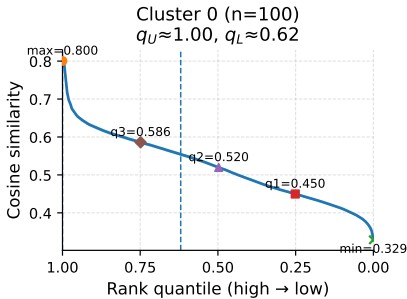

Figure 15: Visualization of clustering analysis of LDAR's zero-shot retrieval behavior on **GPT-4o-mini** for the **BestAnswer** task in the Ada-LEval benchmark (Wang et al., 2024a). We used the LDAR checkpoint pretrained with the Location task for zero-shot evaluation. We collected 100 similarity distributions and clustered them based on LDAR's retrieval band value, In this setting, the analysis yields a single coherent cluster, indicating that LDAR consistently applies a similar retrieval strategy across examples.

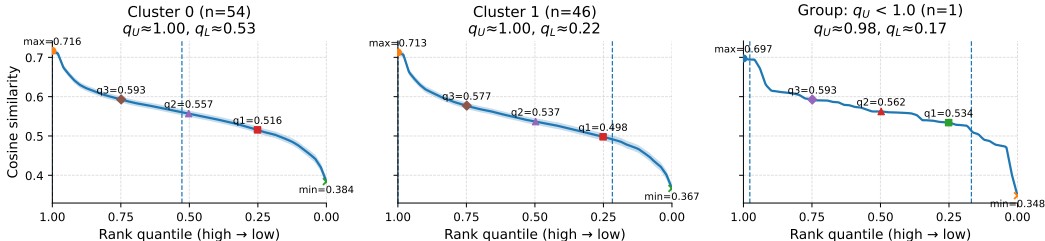

Figure 16: Visualization of clustering analysis of LDAR's zero-shot retrieval behavior on **GPT-4o-mini** for the **HotpotQA** task. We used the LDAR checkpoint pretrained with the Comparison task for zero-shot evaluation. We collected 100 similarity distributions and clustered them based on LDAR's retrieval band value, resulting in two primary clusters (Cluster 0 and Cluster 1). We additionally isolated all instances where the learned upper quantile satisfies $q_U < 1.0$. The figure shows the mean similarity distribution for each cluster along with LDAR's average retrieval band $(q_L, q_U)$. $n$ indicates the number of samples in each cluster and group.

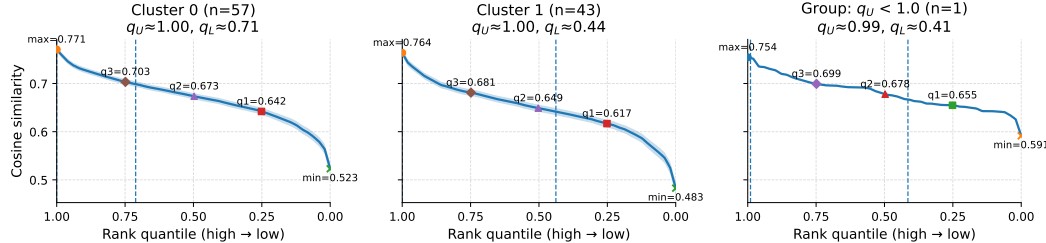

Figure 17: Visualization of clustering analysis of LDAR's zero-shot retrieval behavior on **GPT-4o-mini** for the **NQ** task. We used the LDAR checkpoint pretrained with the Location task for zero-shot evaluation. We collected 100 similarity distributions and clustered them based on LDAR's retrieval band value, resulting in two primary clusters (Cluster 0 and Cluster 1). We additionally isolated all instances where the learned upper quantile satisfies $q_U < 1.0$. The figure shows the mean similarity distribution for each cluster along with LDAR's average retrieval band $(q_L, q_U)$. $n$ indicates the number of samples in each cluster and group.

To show that LDAR's learned retrieval behavior is not just corpus-specific distributional quirks, we additionally conducted a zero-shot evaluation on a long-context benchmark constructed from StackOverflow. In this task, each question is paired with many candidate answers, and the LLM must identify the single answer originally marked as the most helpful (Wang et al., 2024a).

LDAR can zero-shot generalize to tasks with other corpus (i.e., exhibit semantic robustness) as long as the overall similarity distribution is similar in scale. Figure 13 - 17 show that similarity distribution of tasks with different semantics (LaRA (novel, paper, finance), Ada-LEval (StackOverflow), and HELMET (Wikipedia)) are in similar scale, and LDAR succeeds in showing similar behaviors. This also leads to meaningful zero-shot performance on tasks with different semantics (Table 2 and Table 8). Thus, as long as the **embedder** provides a stable and comparable similarity scale between queries and passages across different corpus, the learned LDAR strategy can zero-shot generalize to corpus with different semantics.

Table 8: Zero-shot evaluation on a long-context benchmark constructed from StackOverflow questions paired with many candidate answers, where the LLM must identify the single answer originally marked as the most helpful (Wang et al., 2024a).

| Ada-LEval | LC | RAG | LDAR |
| --- | --- | --- | --- |
| Qwen-2.5-7B | 24.0 (1.0) | 63.0 (0.016) | **65.0** (0.273) |
| Gemini-2.5-pro | 87.5 (1.0) | 80.0 (0.016) | **89.5** (0.669) |

## D.8  ANALYZING LDAR'S LEARNED RETRIEVAL STRATEGY VIA SLIDING WINDOW EXPERIMENTS

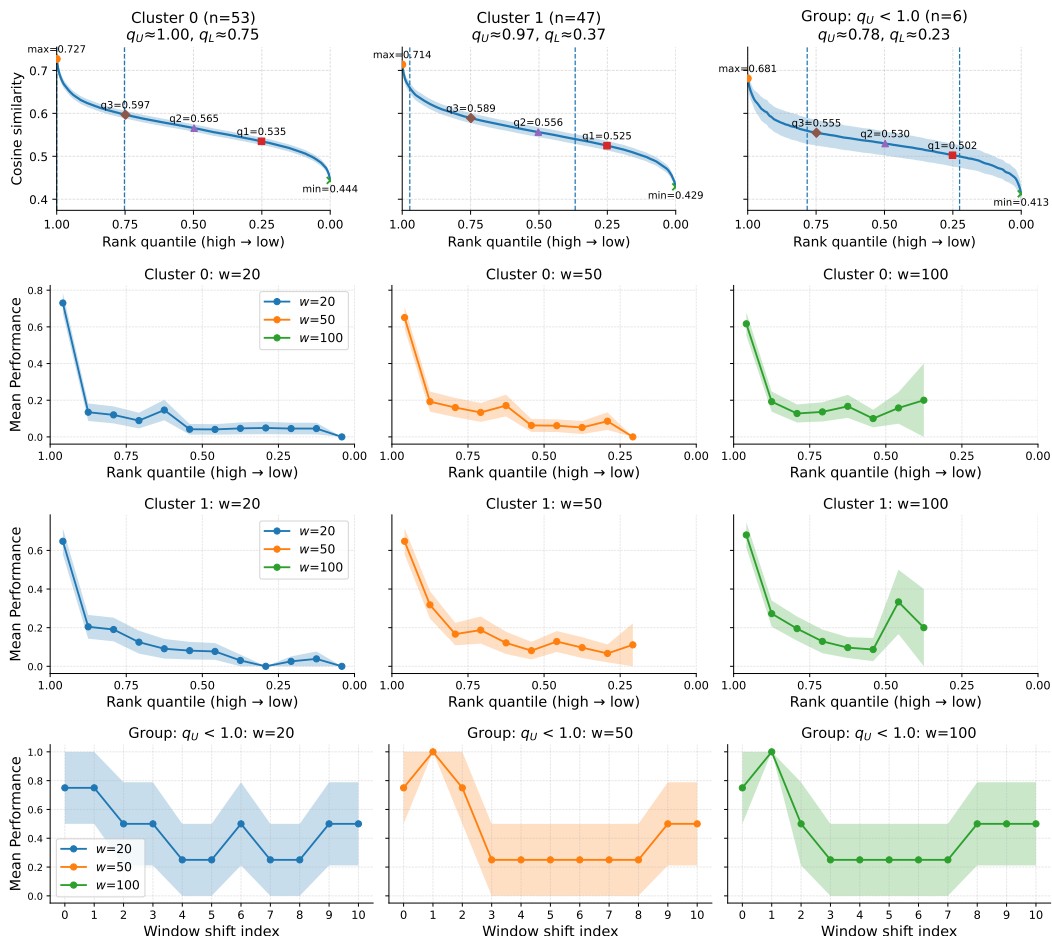

Figure 18: Visualization of clustering analysis over LDAR's retrieval behavior on **Qwen-2.5-7B** in **Location** task, along with the corresponding sliding-window experiments for each cluster. We collected 100 similarity distributions and clustered them based on LDAR's retrieval band value, resulting in two primary clusters (Cluster 0 and Cluster 1). We additionally isolated all instances where the learned upper quantile satisfies $q_U < 1.0$. The figure shows the mean similarity distribution for each cluster along with LDAR's average retrieval band $(q_L, q_U)$. $n$ indicates the number of samples in each cluster and group.

### D.9 ABLATION STUDIES ON ARCHITECTURAL CHOICES

As the number of passages associated with each query varies (Section 4.1), it produces similarity vectors of different lengths. We used Transformer encoder as it can handle variable-length sequences and is able to model relationships within the similarity distribution. To demonstrate the effectiveness of our architectural design choices, we performed an ablation study with different design variants of LDAR. For our MLP variant, we followed the common practice by summarizing each similarity vector into a fixed-size representation using simple pooling (i.e., taking the mean over similarity scores across passages), and feed this pooled representation into a stack of MLP layers (Zaheer et al., 2017). Table 9 shows that LDAR with the transformer encoder shows significantly better performance compared to LDAR with MLP.

Also, we used periodic embeddings because they are highly effective for capturing fine-grained variation in continuous inputs Gorishniy et al. (2022). As shown in Table 9, LDAR with periodic embeddings achieves significantly better performance than LDAR using a standard learnable embedding.

| LLM | Architecture | Location | Reasoning | Comp | Hallu | Average |
|-----|-------------|----------|-----------|------|-------|---------|
| Llama-3.1-8B | Transformer with periodic embedding (LDAR) | **77.6** (0.216) | **59.6** (0.361) | **41.5** (0.501) | **73.3** (0.217) | **63.0** (0.324) |
| Llama-3.1-8B | Transformer with learnable embedding | 77.5 (0.359) | 55.8 (0.158) | 36.5 (0.353) | 67.5 (0.359) | 59.3 (0.307) |
| Llama-3.1-8B | MLP | 71.4 (0.319) | 54.5 (0.286) | 34.1 (0.424) | 70.1 (0.323) | 57.5 (0.338) |
| GPT-4o-mini | Transformer with periodic embedding (LDAR) | **88.8** (0.397) | **80.5** (0.254) | **63.4** (0.613) | **51.8** (0.397) | **71.1** (0.415) |
| GPT-4o-mini | Transformer with learnable embedding | 86.7 (0.520) | 79.2 (0.285) | 58.5 (0.600) | 35.0 (0.521) | 64.9 (0.481) |
| GPT-4o-mini | MLP | 84.6 (0.332) | 77.9 (0.307) | 56.0 (0.497) | 29.8 (0.335) | 63.3 (0.367) |

Table 9: Ablation of architecture choices for LDAR.

### D.10 ABLATION STUDY ON TASK ALIGNMENT

Most tasks in LaRA (such as Location) are single-hop QA tasks, where the answer can be derived from a single relevant passage. In contrast, the Comparison task in LaRA and HotpotQA in HEL-MET both require multi-hop reasoning, where multiple passages must be retrieved and combined to produce the correct answer (Section 5.1).

Our analysis in Section 5.3 shows that for multi-hop tasks, LDAR optimizes to retrieve a larger number of passages compared to other single-hop tasks (e.g., Location task). Based on this observed behavior, we want to highlight that our zero-shot mapping is structurally motivated.

To further validate that the positive zero-shot results are not coincidental, we conducted an additional misaligned transfer experiment. Specifically, we evaluate (1) Location-trained LDAR (single-hop) transferred to HotpotQA (multi-hop), and (2) Comparison-trained LDAR (multi-hop) transferred to NQ (single-hop). As shown in Table 10, misaligned transfers lead to degraded performance: Location → HotpotQA retrieves too few passages to support multi-hop reasoning, while Comparison → NQ retrieves too many passages, introducing additional distraction that harms performance. These results demonstrate that LDAR's zero-shot transfer benefits arise from alignment between task structure, rather than coincidence.

Table 10: Ablation study on task alignment. The top table shows the **aligned task setting** (Comparison → HotpotAQ, Location → NQ). The bottom table shows the **misaligned task setting** (Location → HotpotAQ, Comparison → NQ).

| Task aligned | HotpotQA | | | NQ | | |
|---|---|---|---|---|---|---|
| | LC | RAG | LDAR | LC | RAG | LDAR |
| Qwen-3-4B | 0.56 (1.0) | 0.51 (0.019) | **0.62** (0.536) | 0.51 (1.0) | 0.41 (0.021) | **0.53** (0.486) |
| GPT-4o-mini | 0.64 (1.0) | 0.65 (0.019) | **0.76** (0.629) | **0.59** (1.0) | 0.52 (0.021) | **0.59** (0.374) |

| Task misaligned | HotpotQA | | | NQ | | |
|---|---|---|---|---|---|---|
| | LC | RAG | LDAR | LC | RAG | LDAR |
| Qwen-3-4B | 0.56 (1.0) | 0.51 (0.019) | **0.60** (0.417) | **0.51** (1.0) | 0.41 (0.021) | 0.50 (0.575) |
| GPT-4o-mini | 0.64 (1.0) | 0.65 (0.019) | **0.74** (0.493) | **0.59** (1.0) | 0.52 (0.021) | 0.58 (0.424) |

## D.11 FULL COMPARISON RESULTS OF RETRIEVAL STRATEGIES

In this subsection, we provide more detailed result of Table 1 by showing the performance table for each LLM. Table 11-19 compares the performance of retrieval strategies across context lengths and tasks for each LLM. In this subsection, the columns are grouped by context length. Within each group, **Loc**, **Reas**, **Comp**, and **Hallu** denote the *Location*, *Reasoning*, *Comparison*, and *Hallucination* tasks in LaRA benchmark, respectively.

Table 11: Comparison of retrieval strategies under context length settings of 32k and 128k for **LLaMA-3.1-8B**. Each cell reports accuracy, with the value in parentheses denoting the token-usage ratio relative to LC. The best-performing strategy for each task is highlighted in bold.

| Method | Context Length 32k | | | | | Context Length 128k | | | | |
|---|---|---|---|---|---|---|---|---|---|---|
| | Loc | Reas | Comp | Hallu | Overall | Loc | Reas | Comp | Hallu | Overall |
| Top-1 | 51.7 (0.019) | 17.0 (0.019) | 19.3 (0.018) | **87.3** (0.019) | 43.8 (0.019) | 33.6 (0.005) | 18.1 (0.004) | 2.4 (0.005) | **88.8** (0.005) | 35.7 (0.005) |
| Top-5 | 64.2 (0.095) | 40.4 (0.097) | 45.1 (0.091) | 84.3 (0.095) | 58.5 (0.095) | 64.2 (0.026) | 41.5 (0.024) | 24.3 (0.025) | 77.7 (0.026) | 51.9 (0.025) |
| Top-10 | 71.4 (0.190) | 34.0 (0.194) | 61.2 (0.182) | 83.0 (0.190) | 62.4 (0.189) | 68.3 (0.053) | 50.6 (0.049) | 24.3 (0.050) | 76.7 (0.052) | 55.0 (0.051) |
| Top-25 | 76.7 (0.474) | 42.5 (0.486) | 58.0 (0.457) | 76.0 (0.476) | 63.3 (0.473) | 68.3 (0.133) | 54.5 (0.124) | 24.3 (0.126) | 75.9 (0.131) | 55.8 (0.129) |
| Top-50 | 82.1 (0.866) | 27.6 (0.897) | 51.6 (0.853) | 79.5 (0.862) | 60.2 (0.869) | 67.3 (0.267) | 55.8 (0.249) | 31.7 (0.252) | 72.4 (0.262) | 56.8 (0.258) |
| LC | 82.1 (1.000) | 40.4 (1.000) | 51.6 (1.000) | 78.8 (1.000) | 63.2 (1.000) | 69.3 (1.000) | 50.6 (1.000) | 34.1 (1.000) | 58.4 (1.000) | 53.1 (1.000) |
| RAG | 80.3 (0.095) | 46.8 (0.097) | 58.0 (0.091) | 83.7 (0.095) | 67.2 (0.095) | 72.4 (0.026) | 45.4 (0.024) | 19.5 (0.025) | 78.3 (0.026) | 53.9 (0.025) |
| Self-Route | 78.5 (0.271) | 44.6 (0.385) | 48.3 (0.327) | 75.2 (0.976) | 61.7 (0.490) | 67.3 (0.205) | 53.2 (0.501) | 24.3 (0.933) | 63.2 (0.501) | 52.0 (0.476) |
| Adaptive-$k$ | 58.9 (0.395) | 31.9 (0.362) | 41.9 (0.385) | 84.3 (0.479) | 54.2 (0.405) | 56.1 (0.398) | 37.6 (0.405) | 19.5 (0.675) | 69.8 (0.502) | 45.8 (0.495) |
| BGM | 78.6 (0.047) | 46.8 (0.085) | 54.8 (0.071) | 77.4 (0.056) | 64.4 (0.065) | 69.4 (0.023) | 49.4 (0.018) | 21.9 (0.020) | 77.8 (0.017) | 54.6 (0.019) |
| RankZephyr | 73.2 (0.095) | 42.5 (0.097) | 51.6 (0.091) | 81.3 (0.095) | 62.2 (0.095) | 57.1 (0.026) | 48.0 (0.024) | 9.7 (0.025) | 76.7 (0.026) | 47.9 (0.025) |
| **LDAR** | **91.0** (0.474) | **55.3** (0.545) | **67.7** (0.624) | 82.6 (0.671) | **74.2** (0.579) | **77.6** (0.216) | **59.6** (0.361) | **41.5** (0.501) | 73.3 (0.217) | **63.0** (0.324) |

Table 12: Comparison of retrieval strategies under context length settings of 32k and 128k for **LLaMA-3.2-3B**. Each cell reports accuracy, with the value in parentheses denoting the token-usage ratio relative to LC. The best-performing strategy for each task is highlighted in bold.

| Method | Context Length 32k | | | | | Context Length 128k | | | | |
|---|---|---|---|---|---|---|---|---|---|---|
| | Loc | Reas | Comp | Hallu | Overall | Loc | Reas | Comp | Hallu | Overall |
| Top-1 | 53.5 (0.019) | 14.8 (0.019) | 19.3 (0.018) | **86.5** (0.019) | 43.5 (0.019) | 27.5 (0.005) | 23.3 (0.004) | 2.4 (0.005) | **88.0** (0.005) | 35.3 (0.005) |
| Top-5 | 66.0 (0.095) | 23.4 (0.097) | 35.4 (0.091) | 83.0 (0.095) | 51.9 (0.095) | 56.1 (0.026) | 38.9 (0.024) | 14.6 (0.025) | 77.2 (0.026) | 46.7 (0.025) |
| Top-10 | 71.4 (0.19) | 31.9 (0.194) | 29.0 (0.182) | 83.0 (0.190) | 44.1 (0.189) | 65.3 (0.053) | 42.8 (0.049) | 19.5 (0.05) | 78.5 (0.052) | 51.5 (0.051) |
| Top-25 | 78.5 (0.474) | 27.6 (0.486) | 45.1 (0.457) | 83.4 (0.476) | 58.7 (0.473) | 66.3 (0.133) | 41.5 (0.124) | 17.0 (0.126) | 71.9 (0.131) | 49.2 (0.129) |
| Top-50 | 76.7 (0.866) | 31.9 (0.897) | 38.7 (0.853) | 80.8 (0.862) | 57.0 (0.869) | 66.3 (0.267) | 42.8 (0.249) | 21.9 (0.252) | 63.7 (0.262) | 48.7 (0.258) |
| LC | 80.3 (1.000) | 25.5 (1.000) | 32.2 (1.000) | 65.1 (1.000) | 50.8 (1.000) | 68.3 (1.000) | 33.7 (1.000) | 19.5 (1.000) | 52.0 (1.000) | 43.4 (1.000) |
| RAG | 78.5 (0.095) | 31.9 (0.097) | 22.5 (0.091) | 79.7 (0.095) | 53.2 (0.095) | 70.4 (0.026) | 41.5 (0.024) | 14.6 (0.025) | 69.4 (0.026) | 49.0 (0.025) |
| Self-Route | 78.5 (0.225) | 21.2 (0.116) | 32.2 (0.151) | 68.2 (0.889) | 50.0 (0.345) | 63.2 (0.086) | 35.0 (0.05) | 24.3 (0.263) | 56.0 (0.765) | 44.6 (0.291) |
| Adaptive-$k$ | 58.9 (0.395) | 14.8 (0.362) | 32.2 (0.385) | 78.6 (0.479) | 46.1 (0.405) | 47.9 (0.398) | 24.6 (0.405) | 12.1 (0.675) | 67.4 (0.502) | 38.0 (0.495) |
| BGM | 78.4 (0.058) | 31.2 (0.063) | 45.2 (0.083) | 74.8 (0.037) | 57.4 (0.06) | 66.3 (0.013) | 48.9 (0.008) | 17.0 (0.011) | 75.4 (0.012) | 51.9 (0.011) |
| RankZephyr | 69.6 (0.095) | 19.1 (0.097) | 35.4 (0.091) | 84.7 (0.095) | 52.2 (0.095) | 50.0 (0.026) | 48.0 (0.024) | 12.1 (0.025) | 76.4 (0.026) | 46.6 (0.025) |
| LDAR | **85.7** (0.649) | **34.0** (0.407) | **58.0** (0.776) | 76.5 (0.513) | **63.6** (0.586) | **73.5** (0.15) | **55.5** (0.185) | **34.2** (0.211) | 69.1 (0.15) | **58.1** (0.174) |

Table 13: Comparison of retrieval strategies under context length settings of 32k and 128k for **Qwen-2.5-7B**. Each cell reports accuracy, with the value in parentheses denoting the token-usage ratio relative to LC. The best-performing strategy for each task is highlighted in bold.

| Method | Context Length 32k | | | | | Context Length 128k | | | | |
|---|---|---|---|---|---|---|---|---|---|---|
| | **Loc** | **Reas** | **Comp** | **Hallu** | **Overall** | **Loc** | **Reas** | **Comp** | **Hallu** | **Overall** |
| Top-1 | 53.5 (0.019) | 31.9 (0.019) | 19.3 (0.018) | **88.2** (0.019) | 48.2 (0.019) | 30.6 (0.005) | 24.6 (0.004) | 2.4 (0.005) | **85.7** (0.005) | 35.8 (0.005) |
| Top-5 | 71.4 (0.095) | 31.9 (0.097) | 48.3 (0.091) | 83.0 (0.095) | 58.7 (0.095) | 58.1 (0.026) | 45.4 (0.024) | 21.9 (0.025) | 75.6 (0.026) | 50.2 (0.025) |
| Top-10 | 78.5 (0.19) | 42.5 (0.194) | 58.0 (0.182) | 83.0 (0.190) | 59.7 (0.189) | 66.3 (0.053) | 58.4 (0.049) | 19.5 (0.05) | 71.6 (0.052) | 53.9 (0.051) |
| Top-25 | 85.7 (0.474) | 46.8 (0.486) | 58.0 (0.457) | 81.3 (0.476) | 68.0 (0.473) | 74.4 (0.133) | 58.4 (0.124) | 29.2 (0.126) | 72.2 (0.131) | 58.5 (0.129) |
| Top-50 | 82.1 (0.866) | 46.8 (0.897) | 54.8 (0.853) | 80.8 (0.862) | 66.1 (0.869) | 74.4 (0.267) | 53.2 (0.249) | 34.1 (0.252) | 64.8 (0.262) | 56.6 (0.258) |
| LC | 83.9 (1.000) | 46.8 (1.000) | 54.8 (1.000) | 76.4 (1.000) | 65.5 (1.000) | 35.7 (1.000) | 41.5 (1.000) | 12.2 (1.000) | 42.5 (1.000) | 33.0 (1.000) |
| RAG | 78.5 (0.095) | 44.6 (0.097) | 45.1 (0.091) | 84.3 (0.095) | 63.1 (0.095) | 71.4 (0.026) | 59.7 (0.024) | 24.9 (0.025) | 71.0 (0.026) | 56.8 (0.025) |
| Self-Route | 85.7 (0.272) | 42.5 (0.366) | 41.9 (0.268) | 78.6 (0.972) | 62.2 (0.47) | 61.2 (0.285) | 53.2 (0.227) | 24.3 (0.239) | 67.7 (0.912) | 51.6 (0.416) |
| Adaptive-$k$ | 64.2 (0.395) | 23.4 (0.362) | 48.3 (0.385) | 83.4 (0.479) | 54.8 (0.405) | 37.7 (0.398) | 23.3 (0.405) | 17.0 (0.675) | 70.8 (0.502) | 37.2 (0.495) |
| BGM | 77.0 (0.043) | 49.2 (0.065) | 48.4 (0.066) | 77.4 (0.048) | 63.0 (0.055) | 72.4 (0.011) | 62.3 (0.013) | 39.0 (0.017) | 75.9 (0.012) | 62.4 (0.013) |
| RankZephyr | 75.0 (0.095) | 40.4 (0.097) | 51.6 (0.091) | 82.6 (0.095) | 62.4 (0.095) | 62.2 (0.026) | 55.8 (0.024) | 26.8 (0.025) | 73.2 (0.026) | 54.5 (0.025) |
| LDAR | **91.0** (0.474) | **59.5** (0.39) | **61.2** (0.404) | 83.4 (0.489) | **73.8** (0.439) | **76.5** (0.129) | **64.9** (0.273) | **46.3** (0.307) | 69.1 (0.129) | **64.2** (0.21) |

Table 14: Comparison of retrieval strategies under context length settings of 32k and 128k for **Qwen-3-4B**. Each cell reports accuracy, with the value in parentheses denoting the token-usage ratio relative to LC. The best-performing strategy for each task is highlighted in bold.

| Method | Context Length 32k | | | | | Context Length 128k | | | | |
|---|---|---|---|---|---|---|---|---|---|---|
| | **Loc** | **Reas** | **Comp** | **Hallu** | **Overall** | **Loc** | **Reas** | **Comp** | **Hallu** | **Overall** |
| Top-1 | 53.5 (0.019) | 25.5 (0.019) | 29.0 (0.018) | **93.4** (0.019) | 50.4 (0.019) | 30.6 (0.005) | 31.1 (0.004) | 7.3 (0.005) | **94.1** (0.005) | 40.8 (0.005) |
| Top-5 | 66.0 (0.095) | 48.9 (0.097) | 51.6 (0.091) | 91.7 (0.095) | 64.5 (0.095) | 61.2 (0.026) | 57.1 (0.024) | 31.7 (0.025) | 90.4 (0.026) | 60.1 (0.025) |
| Top-10 | 78.5 (0.19) | 53.1 (0.194) | 51.6 (0.182) | 91.3 (0.19) | 68.6 (0.189) | 68.3 (0.053) | 64.9 (0.049) | 39.0 (0.05) | 87.5 (0.052) | 64.9 (0.051) |
| Top-25 | 82.1 (0.474) | 48.9 (0.486) | 64.5 (0.457) | 90.4 (0.476) | 71.5 (0.473) | 80.6 (0.133) | **68.8** (0.124) | 29.2 (0.126) | 87.0 (0.131) | 66.4 (0.129) |
| Top-50 | 82.1 (0.866) | 46.8 (0.897) | 61.2 (0.853) | 89.5 (0.862) | 69.9 (0.869) | 77.5 (0.267) | 64.9 (0.249) | 43.9 (0.252) | 88.0 (0.262) | 68.6 (0.258) |
| LC | 82.1 (1.000) | 42.5 (1.000) | 61.2 (1.000) | 88.6 (1.000) | 68.6 (1.000) | 79.5 (1.000) | 67.3 (1.000) | 41.4 (1.000) | 82.0 (1.000) | 67.5 (1.000) |
| RAG | 71.4 (0.095) | 44.6 (0.097) | 54.8 (0.091) | 91.3 (0.095) | 65.5 (0.095) | 76.5 (0.026) | 57.1 (0.024) | 36.5 (0.025) | 87.6 (0.026) | 64.4 (0.025) |
| Self-Route | 83.9 (0.239) | 51.0 (0.174) | 58.0 (0.267) | 88.2 (0.945) | 70.3 (0.406) | 75.5 (0.166) | 63.6 (0.151) | 31.7 (0.453) | 81.2 (0.917) | 63.0 (0.422) |
| Adaptive-$k$ | 67.8 (0.395) | 38.2 (0.362) | 45.1 (0.385) | 90.4 (0.479) | 60.4 (0.405) | 60.2 (0.398) | 53.2 (0.405) | 29.2 (0.675) | 88.0 (0.502) | 57.6 (0.495) |
| BGM | 82.3 (0.054) | 49.8 (0.082) | 49.2 (0.072) | 88.7 (0.067) | 67.5 (0.069) | 72.4 (0.023) | 63.6 (0.019) | 41.7 (0.02) | 86.5 (0.022) | 66.0 (0.021) |
| RankZephyr | 69.4 (0.095) | 46.8 (0.097) | 51.6 (0.091) | 90.0 (0.095) | 64.5 (0.095) | 58.1 (0.026) | 66.2 (0.024) | 21.9 (0.025) | 88.6 (0.026) | 58.7 (0.025) |
| LDAR | **87.5** (0.559) | **57.4** (0.486) | **70.9** (0.544) | 90.0 (0.493) | **76.4** (0.52) | **85.7** (0.488) | **68.8** (0.493) | **46.3** (0.537) | 86.0 (0.448) | **71.7** (0.491) |

Table 15: Comparison of retrieval strategies under context length settings of 32k and 128k for **Mistral-Nemo-12B**. Each cell reports accuracy, with the value in parentheses denoting the token-usage ratio relative to LC. The best-performing strategy for each task is highlighted in bold.

| Method | Context Length 32k | | | | | Context Length 128k | | | | |
|---|---|---|---|---|---|---|---|---|---|---|
| | Loc | Reas | Comp | Hallu | Overall | Loc | Reas | Comp | Hallu | Overall |
| Top-1 | 51.7 (0.019) | 27.6 (0.019) | 16.1 (0.018) | **74.7** (0.019) | 42.5 (0.019) | 33.6 (0.005) | 33.7 (0.004) | 9.7 (0.005) | **67.9** (0.005) | 36.2 (0.005) |
| Top-5 | 66.0 (0.095) | 46.8 (0.097) | 45.1 (0.091) | 70.8 (0.095) | 57.2 (0.095) | 61.2 (0.026) | 54.5 (0.024) | 29.2 (0.025) | 50.0 (0.026) | 48.7 (0.025) |
| Top-10 | 76.7 (0.19) | 46.8 (0.194) | **58.0** (0.182) | 55.6 (0.19) | 59.3 (0.189) | 66.3 (0.053) | 51.9 (0.049) | 43.9 (0.05) | 37.5 (0.052) | 49.9 (0.051) |
| Top-25 | 67.8 (0.474) | 34.0 (0.486) | 25.8 (0.457) | 41.3 (0.476) | 42.2 (0.473) | 67.3 (0.133) | 48.0 (0.124) | 24.3 (0.126) | 18.5 (0.131) | 39.5 (0.129) |
| Top-50 | 67.8 (0.866) | 34.0 (0.897) | 41.9 (0.853) | 30.0 (0.862) | 43.4 (0.869) | 54.0 (0.267) | 45.4 (0.249) | 36.5 (0.252) | 17.4 (0.262) | 38.3 (0.258) |
| LC | 73.2 (1.000) | 29.7 (1.000) | 38.7 (1.000) | 39.5 (1.000) | 45.3 (1.000) | 28.6 (1.000) | 32.5 (1.000) | 12.2 (1.000) | 12.7 (1.000) | 21.5 (1.000) |
| RAG | 73.2 (0.095) | 51.0 (0.097) | 54.8 (0.091) | 68.6 (0.095) | 61.9 (0.095) | 67.3 (0.026) | 45.4 (0.024) | 36.5 (0.025) | 48.4 (0.026) | 49.4 (0.025) |
| Self-Route | 76.7 (0.271) | 42.5 (0.252) | 54.8 (0.208) | 39.5 (0.964) | 53.4 (0.424) | 62.2 (0.195) | 51.9 (0.214) | 21.9 (0.453) | 14.0 (0.917) | 37.5 (0.445) |
| Adaptive-$k$ | 55.3 (0.395) | 19.1 (0.362) | 48.3 (0.385) | 50.0 (0.479) | 43.2 (0.405) | 36.7 (0.398) | 22.0 (0.405) | 14.6 (0.675) | 34.1 (0.502) | 26.9 (0.495) |
| BGM | 78.5 (0.037) | 54.6 (0.042) | 51.9 (0.04) | 59.1 (0.038) | 61.0 (0.039) | 64.3 (0.017) | 58.4 (0.006) | 31.7 (0.006) | 50.0 (0.015) | 51.1 (0.011) |
| RankZephyr | 75.0 (0.095) | 36.1 (0.097) | 32.2 (0.091) | 63.9 (0.095) | 51.8 (0.095) | 53.0 (0.026) | 54.5 (0.024) | 34.1 (0.025) | 51.8 (0.026) | **73.3** (0.025) |
| LDAR | **83.9** (0.236) | **57.4** (0.172) | **58.0** (0.245) | 50.0 (0.208) | **62.3** (0.215) | **73.5** (0.102) | **59.7** (0.048) | **46.3** (0.006) | 24.1 (0.102) | 50.9 (0.065) |

Table 16: Comparison of retrieval strategies under context length settings of 32k and 128k for **GPT-4o**. Each cell reports accuracy, with the value in parentheses denoting the token-usage ratio relative to LC. The best-performing strategy for each task is highlighted in bold.

| Method | Context Length 32k | | | | | Context Length 128k | | | | |
|---|---|---|---|---|---|---|---|---|---|---|
| | Loc | Reas | Comp | Hallu | Overall | Loc | Reas | Comp | Hallu | Overall |
| Top-1 | 58.9 (0.019) | 42.5 (0.019) | 35.4 (0.018) | **90.4** (0.019) | 56.8 (0.019) | 33.6 (0.005) | 32.4 (0.004) | 17.0 (0.005) | **87.8** (0.005) | 42.7 (0.005) |
| Top-5 | 78.5 (0.095) | 70.2 (0.097) | 58.0 (0.091) | 83.0 (0.095) | 72.4 (0.095) | 65.3 (0.026) | 70.1 (0.024) | 31.7 (0.25) | 74.8 (0.026) | 60.5 (0.025) |
| Top-10 | 83.9 (0.19) | 63.8 (0.194) | 67.7 (0.182) | 78.6 (0.19) | 73.5 (0.189) | 73.4 (0.053) | 74.0 (0.049) | 43.9 (0.05) | 66.1 (0.052) | 64.3 (0.051) |
| Top-25 | 91.0 (0.474) | 61.7 (0.486) | 77.4 (0.457) | 75.2 (0.476) | 76.3 (0.473) | 82.6 (0.133) | 79.2 (0.124) | 51.2 (0.126) | 63.4 (0.131) | 69.1 (0.129) |
| Top-50 | 92.8 (0.866) | 70.2 (0.897) | 77.4 (0.854) | 72.1 (0.862) | 78.1 (0.87) | 79.5 (0.267) | 77.9 (0.249) | 65.8 (0.252) | 61.9 (0.262) | 71.3 (0.258) |
| LC | 87.5 (1.000) | 70.2 (1.000) | 80.6 (1.000) | 72.3 (1.000) | 77.6 (1.000) | 90.8 (1.000) | 80.5 (1.000) | 65.8 (1.000) | 56.3 (1.000) | 73.4 (1.000) |
| RAG | 83.9 (0.095) | 55.3 (0.097) | 67.7 (0.091) | 78.3 (0.095) | 71.3 (0.095) | 80.6 (0.026) | 63.6 (0.024) | 46.3 (0.25) | 67.3 (0.026) | 64.5 (0.082) |
| Self-Route | 91.0 (0.302) | 70.2 (0.271) | 70.9 (0.327) | 75.6 (0.984) | 76.9 (0.471) | 80.6 (0.295) | 80.5 (0.316) | 63.4 (0.762) | 60.5 (0.956) | 71.2 (0.582) |
| Adaptive-$k$ | 75.0 (0.395) | 59.5 (0.362) | 64.5 (0.385) | 81.3 (0.479) | 70.1 (0.405) | 64.2 (0.398) | 57.1 (0.405) | 48.7 (0.675) | 67.4 (0.502) | 59.4 (0.495) |
| BGM | 80.4 (0.045) | 61.7 (0.081) | 61.3 (0.058) | 73.9 (0.038) | 69.3 (0.056) | 75.5 (0.017) | 66.2 (0.02) | 31.7 (0.018) | 70.4 (0.018) | 60.9 (0.017) |
| RankZephyr | 83.9 (0.095) | 61.7 (0.097) | 64.5 (0.091) | 82.1 (0.095) | 73.1 (0.095) | 64.2 (0.026) | 75.3 (0.024) | 36.5 (0.025) | 71.9 (0.026) | 62.0 (0.025) |
| LDAR | **94.6** (0.597) | **72.3** (0.73) | **87.0** (0.514) | 76.9 (0.598) | **82.7** (0.61) | **91.8** (0.376) | **84.4** (0.35) | **68.3** (0.58) | 60.3 (0.376) | **76.2** (0.42) |

Table 17: Comparison of retrieval strategies under context length settings of 32k and 128k for **GPT-4o-mini**. Each cell reports accuracy, with the value in parentheses denoting the token-usage ratio relative to LC. The best-performing strategy for each task is highlighted in bold.

| Method | *Context Length 32k* | | | | | *Context Length 128k* | | | | |
|---|---|---|---|---|---|---|---|---|---|---|
| | Loc | Reas | Comp | Hallu | Overall | Loc | Reas | Comp | Hallu | Overall |
| Top-1 | 55.3 (0.019) | 36.1 (0.019) | 41.9 (0.018) | **81.3** (0.019) | 53.6 (0.019) | 32.6 (0.005) | 40.2 (0.004) | 21.9 (0.005) | **76.4** (0.005) | 42.8 (0.005) |
| Top-5 | 78.5 (0.095) | 48.9 (0.097) | 74.1 (0.091) | 75.6 (0.095) | 69.3 (0.095) | 63.2 (0.026) | 66.2 (0.024) | 41.4 (0.025) | 62.6 (0.026) | 58.4 (0.025) |
| Top-10 | 80.3 (0.19) | 51.0 (0.194) | 74.1 (0.182) | 70.0 (0.19) | 68.9 (0.189) | 69.3 (0.053) | 67.5 (0.049) | 51.2 (0.05) | 53.1 (0.052) | 60.3 (0.051) |
| Top-25 | 85.7 (0.474) | 61.7 (0.486) | 77.4 (0.457) | 64.3 (0.476) | 72.3 (0.473) | 77.5 (0.133) | 75.3 (0.124) | 43.9 (0.126) | 45.2 (0.131) | 60.5 (0.129) |
| Top-50 | **89.2** (0.866) | 57.4 (0.897) | 77.4 (0.853) | 62.1 (0.862) | 71.5 (0.869) | 79.5 (0.267) | 74.0 (0.249) | 48.7 (0.252) | 40.7 (0.262) | 60.7 (0.258) |
| LC | 87.5 (1.000) | 55.3 (1.000) | 77.4 (1.000) | 52.8 (1.000) | 68.2 (1.000) | 81.6 (1.000) | 71.4 (1.000) | 58.3 (1.000) | 32.0 (1.000) | 60.8 (1.000) |
| RAG | 82.1 (0.095) | 59.5 (0.097) | 70.9 (0.091) | 67.1 (0.095) | 69.9 (0.095) | 70.4 (0.026) | 72.7 (0.024) | 46.3 (0.025) | 59.3 (0.026) | 62.2 (0.025) |
| Self-Route | 87.5 (0.336) | 51.0 (0.213) | 74.1 (0.268) | 61.7 (0.968) | 68.6 (0.446) | 71.4 (0.195) | 72.7 (0.151) | 43.9 (0.524) | 29.3 (0.946) | 54.3 (0.454) |
| Adaptive-$k$ | 75.0 (0.395) | 55.3 (0.362) | 64.5 (0.385) | 70.4 (0.479) | 66.3 (0.405) | 59.1 (0.398) | 50.6 (0.405) | 41.4 (0.675) | 49.4 (0.502) | 50.1 (0.495) |
| BGM | 82.1 (0.052) | 57.4 (0.063) | 80.3 (0.082) | 64.3 (0.056) | 71.0 (0.063) | 78.6 (0.016) | 67.3 (0.01) | 36.6 (0.02) | 57.9 (0.012) | 60.1 (0.014) |
| RankZephyr | 83.9 (0.095) | 53.1 (0.097) | 74.1 (0.091) | 70.0 (0.095) | 70.3 (0.095) | 62.2 (0.026) | 66.2 (0.024) | 31.7 (0.025) | 58.9 (0.026) | 54.8 (0.025) |
| LDAR | **89.2** (0.666) | **70.2** (0.433) | **80.6** (0.741) | 60.4 (0.684) | **75.1** (0.631) | **88.8** (0.397) | **80.5** (0.254) | **63.4** (0.613) | 51.8 (0.397) | **71.1** (0.415) |

Table 18: Comparison of retrieval strategies under context length settings of 32k and 128k for **Gemini-2.5-pro**. Each cell reports accuracy, with the value in parentheses denoting the token-usage ratio relative to LC. The best-performing strategy for each task is highlighted in bold.

| Method | *Context Length 32k* | | | | | *Context Length 128k* | | | | |
|---|---|---|---|---|---|---|---|---|---|---|
| | Loc | Reas | Comp | Hallu | Overall | Loc | Reas | Comp | Hallu | Overall |
| Top-1 | 57.1 (0.019) | 31.9 (0.019) | 29.0 (0.018) | **93.4** (0.019) | 52.8 (0.019) | 29.5 (0.005) | 33.7 (0.004) | 4.8 (0.005) | **93.3** (0.005) | 40.3 (0.005) |
| Top-5 | 76.7 (0.095) | 59.5 (0.097) | 58.0 (0.091) | 90.8 (0.095) | 71.2 (0.095) | 61.2 (0.026) | 58.4 (0.024) | 29.2 (0.025) | 85.4 (0.026) | 58.6 (0.025) |
| Top-10 | 83.9 (0.19) | 63.8 (0.194) | 58.0 (0.182) | 89.5 (0.19) | 73.8 (0.189) | 75.5 (0.053) | 64.9 (0.049) | 36.5 (0.05) | 82.5 (0.052) | 64.8 (0.051) |
| Top-25 | 85.7 (0.474) | 63.8 (0.486) | 64.5 (0.457) | 86.5 (0.476) | 75.1 (0.473) | 81.6 (0.133) | 76.6 (0.124) | 53.6 (0.126) | 78.0 (0.131) | 72.4 (0.129) |
| Top-50 | 83.9 (0.866) | 63.8 (0.897) | **67.7** (0.853) | 84.7 (0.862) | 75.0 (0.869) | 81.6 (0.267) | 76.6 (0.249) | 43.9 (0.252) | 71.9 (0.262) | 68.5 (0.258) |
| LC | 89.2 (1.000) | 59.5 (1.000) | 62.1 (1.000) | 86.5 (1.000) | 74.3 (1.000) | **91.8** (1.000) | 80.5 (1.000) | 46.3 (1.000) | 76.6 (1.000) | 73.8 (1.000) |
| RAG | 82.1 (0.095) | 61.7 (0.097) | 61.2 (0.091) | 87.3 (0.095) | 73.1 (0.095) | 75.5 (0.026) | 64.9 (0.024) | 36.5 (0.025) | 84.4 (0.026) | 65.3 (0.025) |
| Self-Route | 89.2 (0.255) | 63.8 (0.291) | 64.5 (0.414) | 86.0 (0.968) | 75.9 (0.482) | 86.7 (0.285) | 71.4 (0.239) | 43.9 (0.596) | 70.1 (0.935) | 68.0 (0.514) |
| Adaptive-$k$ | 69.6 (0.395) | 44.6 (0.362) | 61.2 (0.385) | 90.0 (0.479) | 66.3 (0.405) | 70.4 (0.398) | 53.2 (0.405) | 41.4 (0.675) | 78.3 (0.502) | 60.8 (0.495) |
| BGM | 83.9 (0.068) | 61.7 (0.07) | 54.8 (0.067) | 80.3 (0.049) | 70.2 (0.064) | 68.3 (0.023) | 64.9 (0.019) | 29.3 (0.022) | 81.3 (0.021) | 60.9 (0.021) |
| RankZephyr | 82.1 (0.095) | 65.9 (0.097) | 58.0 (0.091) | 90.4 (0.095) | 74.1 (0.095) | 64.2 (0.026) | 71.4 (0.024) | 21.9 (0.025) | 87.8 (0.026) | 61.3 (0.025) |
| LDAR | **92.8** (0.641) | **68.0** (0.668) | **67.7** (0.668) | 86.9 (0.641) | **78.8** (0.655) | **91.8** (0.738) | **83.1** (0.52) | **63.4** (0.632) | 80.7 (0.738) | **79.8** (0.657) |

Table 19: Comparison of retrieval strategies under context length settings of 32k and 128k for **Gemini-2.5-flash**. Each cell reports accuracy, with the value in parentheses denoting the token-usage ratio relative to LC. The best-performing strategy for each task is highlighted in bold.

| Method | Context Length 32k | | | | | Context Length 128k | | | | |
|---|---|---|---|---|---|---|---|---|---|---|
| | **Loc** | **Reas** | **Comp** | **Hallu** | **Overall** | **Loc** | **Reas** | **Comp** | **Hallu** | **Overall** |
| Top-1 | 55.3 (0.019) | 38.2 (0.019) | 25.8 (0.018) | **92.6** (0.019) | 53.0 (0.019) | 30.6 (0.005) | 28.5 (0.004) | 4.8 (0.005) | **93.3** (0.005) | 39.3 (0.005) |
| Top-5 | 78.5 (0.095) | 68.0 (0.097) | 61.2 (0.091) | 87.8 (0.095) | 73.9 (0.095) | 57.1 (0.026) | 55.8 (0.024) | 34.1 (0.025) | 85.4 (0.026) | 58.1 (0.025) |
| Top-10 | 85.7 (0.19) | 59.5 (0.194) | 61.2 (0.182) | 85.2 (0.19) | 72.9 (0.189) | 65.3 (0.053) | 63.6 (0.049) | 39.0 (0.05) | 82.5 (0.052) | 62.6 (0.051) |
| Top-25 | 87.5 (0.474) | 59.5 (0.486) | 64.5 (0.457) | 82.1 (0.476) | 73.4 (0.473) | 78.5 (0.133) | 70.1 (0.124) | 43.9 (0.126) | 78.0 (0.131) | 67.6 (0.129) |
| Top-50 | 82.1 (0.866) | 63.8 (0.897) | 54.8 (0.853) | 80.0 (0.862) | 70.2 (0.869) | 80.6 (0.267) | 68.8 (0.249) | 51.2 (0.252) | 71.9 (0.262) | 68.1 (0.258) |
| LC | 85.7 (1.000) | 63.8 (1.000) | 74.1 (1.000) | 80.4 (1.000) | 76.0 (1.000) | 87.8 (1.000) | 75.3 (1.000) | 56.1 (1.000) | 68.8 (1.000) | 72.0 (1.000) |
| RAG | 82.1 (0.095) | 59.5 (0.097) | 45.1 (0.091) | 86.5 (0.095) | 68.3 (0.095) | 74.5 (0.026) | 62.3 (0.024) | 41.4 (0.025) | 85.7 (0.026) | 66.0 (0.025) |
| Self-Route | **91.0** (0.287) | 65.9 (0.232) | 61.2 (0.239) | 80.8 (0.948) | 74.7 (0.426) | 81.6 (0.355) | 74.0 (0.29) | 58.5 (0.619) | 70.1 (0.935) | 71.0 (0.55) |
| Adaptive-$k$ | 66.0 (0.395) | 46.8 (0.362) | 48.3 (0.385) | 87.3 (0.479) | 62.1 (0.405) | 64.2 (0.398) | 55.8 (0.405) | 46.3 (0.675) | 78.3 (0.502) | 61.2 (0.495) |
| BGM | 83.9 (0.064) | 55.3 (0.08) | 48.4 (0.05) | 84.3 (0.038) | 68.0 (0.058) | 66.3 (0.02) | 68.8 (0.018) | 39.0 (0.02) | 84.1 (0.016) | 64.5 (0.018) |
| RankZephyr | 78.5 (0.095) | 57.4 (0.097) | 51.6 (0.091) | 88.2 (0.095) | 68.9 (0.095) | 60.2 (0.026) | 72.7 (0.024) | 21.9 (0.025) | 84.4 (0.026) | 59.8 (0.025) |
| LDAR | **91.0** (0.611) | **70.2** (0.716) | **80.6** (0.556) | 83.0 (0.61) | **81.2** (0.623) | **89.8** (0.498) | **83.1** (0.654) | **68.3** (0.602) | 71.1 (0.565) | **78.1** (0.58) |

