# OpenReview forum: "Beyond RAG vs. Long-Context: Learning Distraction-Aware Retrieval for Efficient Knowledge Grounding"
_ICLR.cc/2026/Conference — ICLR 2026 Poster_

### Official Review · Reviewer_VnKc · 2025-10-28

**Soundness:** 3
**Presentation:** 3
**Contribution:** 3
**Rating:** 8
**Confidence:** 3

**Summary:**

This paper address the distracting issue caused by RAG. They propose a method called LDAR which learns a lightweight retriever that learns to select passages to minimize potential interference from distracting passage. Their retriver operates on the cosine similarity distribution and selects a dynamic set of passages from a contiguous quantile interval to balance coverage and distraction. For example, if the risk of having distraction is higher, retrieves passages from a narrow quantile interval, minimizing the risk of having distraction. The retriever is trained with the reward signal of whether the downstream LLM gives a correct answer using the retrieved passages. Their expeirments on various benchmarks show significantly better results than the baseline methods.

**Strengths:**

1. The motivation of this paper is solid, learning to balance coverage and distraction for RAG is an important question.
2. The proposed method is lightweight by only finetuning a retriever model.
3. The experimental setups are comprehensive, the zero-shot experiment is persuasive and the performance is good.

**Weaknesses:**

1. An analysis of the proposed method is needed. For example, it would be interesting to compare with a baseline that learns to rerank the retrieved passages alone with the same optimization objective.
2. The interpretability of the addressed distaction issue is limited. It would be interesting to see how the proposed retriever balance coverage and distraction, some case studies would be helpful.

**Questions:**

1. what is the passages ratios compared with the initial top-N because LDAR quantile cutting step.
2. For more capable or larger LLMs, is the quantile cutting still useful compare with using long-context.

---

> ### Author Response · Authors · 2025-11-27
>
> Thank you for your valuable feedback. Please find the responses to your questions below.
>
> **Q1: The interpretability of the addressed distraction issue is limited. It would be interesting to see how the proposed retriever balance coverage and distraction, some case studies would be helpful.**
>
> Thank you for the thoughtful comment. **Figure 8 in Appendix D.2** presents a detailed case study of LDAR’s retrieval behavior, complementing the example shown in Figure 3 of the main paper.
>
> When the similarity distribution shows a clear high-similarity region, LDAR tends to focus narrowly on that region. In these cases, the useful passage tends to be located within the top rank passages as illustrated in Figure 8 (top).
>
> Conversely, when the overall similarity distribution is low in value and relatively flat in shape, LDAR expands its retrieval interval to ensure broader coverage, even at the risk of including more distracting passages. In these cases, the useful passage is often not located within the top rank passages as illustrated in Figure 8 (bottom), making a wider retrieval band necessary.
>
> Although LDAR does not receive any textual information (Line 181), this case study demonstrates that the shape of the similarity distribution is indicative of where useful passages tend to appear. LDAR learns these patterns and adapts its retrieval strategy accordingly.
>
> **Q2: An analysis of the proposed method is needed. For example, it would be interesting to compare with a baseline that learns to rerank the retrieved passages alone with the same optimization objective.**
>
> Thank you for your feedback. BGM serves as precisely such a baseline as it reranks the top-k retrieved passages and is trained with the same optimization objective as LDAR. Although BGM is effective in the 32k context-length setting, its performance gains diminish substantially in long-context scenarios (128k) (Table 1). Increasing k may appear to be a straightforward way to mitigate this issue, but doing so dramatically expands the combinatorial subset-selection space, which in practice leads the model to converge to suboptimal ranking strategies and limits overall performance (Section 5.4).
>
> **Q3: What is the passages ratios compared with the initial top-N because LDAR quantile cutting step.**
>
> Thank you for your comment. In Table 1, the values in parentheses represent the token-usage ratio relative to the Long-Context (LC) setting (Line 327). Since LC corresponds to retrieving all passages, this ratio provides a consistent and comparable measure of how many passages each method retrieves. For example, in the Location task under the 128k setting, LDAR retrieves 0.209 * 128,000 = 26,752 tokens, whereas LC retrieves the full 128,000 tokens (1.0* 128,000 = 128,000), and Top-5 retrieves 0.026*128,000 = 3,328 tokens.
>
> In addition, we followed the standard LaRA benchmark [1] chunking procedure, forming 600-token passages with 100-token overlap. As the size of each passage is equal in length, we can compute the number of passages directly from the information provided in Table 1. This results in approximately **64 passages** for the 32k context-length setting and **256 passages** for the 128k setting, which are used consistently across all methods. Since the reported token-usage ratio corresponds exactly to the fraction of passages retrieved, the number of passages retrieved by each baseline can be computed directly from the table.
>
> **Q4: For more capable or larger LLMs, is the quantile cutting still useful compare with using long-context.**
>
> Thank you for your suggestion. The recent release of Gemini-3-Pro provides an excellent opportunity to evaluate whether LDAR continues to offer benefits when paired with a more capable long-context model. However, due to an ongoing API-access issue with Gemini-3-Pro, we were unable to complete these experiments in time for the current version of the manuscript. We plan to include a comprehensive evaluation with Gemini-3-Pro in a future version of the manuscript.
>
> [1] Li, Kuan, et al. "LaRA: Benchmarking Retrieval-Augmented Generation and Long-Context LLMs--No Silver Bullet for LC or RAG Routing." arXiv preprint arXiv:2502.09977 (2025).

---

### Official Review · Reviewer_brC4 · 2025-10-30

**Soundness:** 3
**Presentation:** 3
**Contribution:** 2
**Rating:** 6
**Confidence:** 3

**Summary:**

This paper presents LDAR(Learning Distraction-Aware Retrieval), a method for adaptive retrieval that compellingly addresses the performance-cost tradeoff between RAG and full long-context. The core contribution which involves identifying the problem of LLMs getting incorrect results given both gold passage and distraction passage, and a lightweight, band-based retriever that is sound and justified well againts the "distraction problem"(Figure 3 , Section 4.1). The experimental evaluation is thorough, demonstrating LDAR's effectiveness and adaptability across a wide range of modern LLMs and challenging tasks. However, the method's design, which feeds raw similarity scores to a Transformer , raises significant concerns about overfitting to task-specific patterns, a limitation the authors themselves acknowledge. This casts doubt on the true generalizability claimed in the zero-shot experiments (see Section 5.5 ). Other key weaknesses include lack of ablation studies and justification for key architectural choices the complete omission of the method's training cost.

**Strengths:**

## Originality
The paper's primary originality lies in its precise and insightful problem formulation. It moves beyond the simplistic "RAG vs. Long-Context" debate to investigate why RAG systems fail. The core insight, clearly demonstrated in Section 3 and Figure 2, is that performance degradation is not just from irrelevant passages but that even a set of exclusively gold passages can collectively confuse an LLM and lead to incorrect answers. Identifying this "distraction" effect, even among relevant documents, is a novel and important contribution.

The proposed solution, LDAR, is also original. Instead of relying on complex textual rerankers or fine-tuning the LLM, it introduces a lightweight, "score-only" adaptive retriever. The "band-based" retrieval strategy (Section 4.1) is a creative mechanism to make the reinforcement learning problem tractable, cleverly reducing the search space from a complex combinatorial problem (as in Bernoulli sampling) to a low-dimensional, continuous one.

## Quality
The empirical quality of this work is a major strength. The authors validate LDAR with a thorough and rigorous experimental setup (Section 5).

1. Comprehensive Benchmarking: The method is tested across a diverse and modern set of both open-source (Llama-3.1, Qwen-2.5, Mistral-Nemo) and closed-source (GPT-4o, Gemini-2.5) models, demonstrating broad applicability.

2. Challenging Tasks: The evaluation spans six knowledge-intensive benchmarks, including the difficult LaRA tasks (Location, Reasoning, Comparison) and HELMET's long-context adaptations of HotpotQA and NQ.

3. Strong Results: The results in Table 1 are compelling. LDAR consistently achieves a superior performance-to-cost ratio, significantly outperforming the full long-context (LC) baseline in overall accuracy while using substantially fewer tokens (often achieving ~2x token efficiency).

4. Well-Justified Method: The core design choice of "band-based" retrieval is well-justified empirically by the comparison in Figure 3, which clearly shows it finds a more stable and effective trade-off than the Bernoulli-based alternative.

## Clarity
The paper is exceptionally well-written, organized, and easy to follow, earning a top score for presentation. The authors use visualizations effectively to build a clear and logical narrative.

## Significance
This paper addresses a highly significant and practical problem for the LLM community. As context windows expand to 1M tokens, the question of how to best utilize this vast context for optimal cost-performance is of paramount importance. This work provides a practical and effective solution that moves "beyond" the binary choice of RAG vs. full context.

**Weaknesses:**

## Serious Concerns Regarding Generalization and Overfitting
- The model's score-only policy may overfit similarity-shape statistics of a corpus. LDAR uses only the cosine-similarity distribution (no text), and explicitly restricts access to textual content for scalability. While elegant, this design risks learning corpus-specific distributional quirks rather than semantic robustness, limiting the proposed method's contribution.

- Zero-shot section (Sec. 5.5) lacks mechanism analysis. Authors transfer LaRA-trained policies to HELMET (HotpotQA/NQ) and report modest gains, but do not explain why a Comparison-trained policy improves HotpotQA or why a Location-trained policy improves NQ. Given that different tasks have vastly different retrieval needs, this positive result may be a coincidence.

- The "score-only" architecture is also vulnerable to overfitting to the statistical properties of the passage database (e.g., LaRA's mix of novels and finance reports). The paper provides no evidence that a policy learned on this database distribution would generalize to a database with a different profile (e.g., social media, legal documents). The test on HELMET's Wikipedia database (Section 5.5) is not a sufficient stress test and lacks a deep analysis of the distribution shift.

## Missing Ablation Studies for Key Architectural Choices
- The paper employs a Transformer Encoder to process the sequence of similarity scores. This is a non-trivial architectural choice, implying that the relationships between scores are important. However, this is not justified with a baseline, such as a simpler Multi-Layer Perceptron (MLP), to prove this complexity is necessary.
- The model uses "Periodic Embedding" to encode the raw scalar similarity scores. This is a specific and non-obvious design choice. The paper provides no justification or ablation study comparing this to simpler, standard alternatives (such as learnable embedding with truncation of sorted scores).

## Omission of Training Cost and Practicality
- The paper's central claim is improved inference efficiency (i.e., lower token usage). However, it provides no information whatsoever about the training cost. The RL training loop, as formulated in Eq. (3), requires a forward pass of the (potentially massive) pretrained LLM $F_{\psi}$ to obtain the reward signal $r_{\psi}$ for every gradient step. This suggests the training process may be prohibitively expensive.
- The paper provides no data on the number of LLM calls required for convergence or the total wall-clock training time. This omission makes it impossible to assess the method's practical viability or cost-benefit trade-off.

**Questions:**

Please list up and carefuly describe any questins and suggestions for the authors.Think ofthe things where a response from the author can change youropinion, clarify a confusion or address a limitation. This is important for a productive rebuttal and discussion phase with the authors.

- Can you extend Sec. 5.5 with broader zero-shot tests? Add more tasks (train/test task cross-matrix) and more corpus database.
- Have you done any ablation on the necessity of choosing transformers as the backbone of the retriever?
- Could you provide a clear training cost-benifit analysis?

---

> ### Author Response · Authors · 2025-11-27
>
> Thank you for your review and constructive suggestions. We address your questions below.
>
> **Q1: The "score-only" architecture is also vulnerable to overfitting to the statistical properties of the passage database (e.g., LaRA's mix of novels and finance reports). The paper provides no evidence that a policy learned on this database distribution would generalize to a database with a different profile (e.g., social media, legal documents). The test on HELMET's Wikipedia database (Section 5.5) is not a sufficient stress test and lacks a deep analysis of the distribution shift.**
>
> **&**
>
> **Q2: The model's score-only policy may overfit similarity-shape statistics of a corpus. LDAR uses only the cosine-similarity distribution (no text), and explicitly restricts access to textual content for scalability. While elegant, this design risks learning corpus-specific distributional quirks rather than semantic robustness, limiting the proposed method's contribution.**
>
> Thank you for your thoughtful feedback. To show that LDAR’s learned retrieval behavior is not just corpus-specific distributional quirks, we have done a case study of LDAR’s retrieval behavior in **Figure 8 in Appendix D.2**. Additionally, we conducted a zero-shot evaluation on a long-context benchmark constructed from StackOverflow questions paired with many candidate answers, where the LLM must identify the single answer originally marked as the most helpful [1].
>
> Figure 8 (top) demonstrates that when the similarity distribution shows a clear high-similarity region, LDAR tends to focus narrowly on that region. In these cases, the useful passage tends to be located within the top rank passages.
>
> Conversely, when the overall similarity distribution is low in value and relatively flat in shape, LDAR expands its retrieval interval to ensure broader coverage, even at the risk of including more distracting passages. In these cases, the useful passage is often not located within the top rank passages as illustrated in Figure 8 (bottom), making a wider retrieval band necessary.
>
> Although LDAR does not receive any textual information (Line 181), this case study demonstrates that the shape of the similarity distribution is predictive of where useful passages tend to appear.
>
> Thus, we want to highlight that LDAR can zero-shot generalize to tasks with other corpus (i.e., exhibit semantic robustness) as long as the overall similarity distribution is similar in scale. **Figure 13-17 in Appendix D.6 and D.7** shows that similarity distribution of tasks with different semantics (LaRA (novel, paper, finance), Ada-LEval (StackOverflow), and HELMET (Wikipedia)) are in similar scale, and LDAR succeeds in showing similar behaviors. This also leads to moderate zero-shot performance on tasks with different semantics (Table 2 and Table below). Thus, as long as the **embedder** provides a stable and comparable similarity scale between queries and passages across different corpus, the learned LDAR policy can zero-shot generalize to corpus with different semantics.
>
> | Ada-LEval         | LC             | RAG            | LDAR  |
> |----------------|----------------|----------------|----------------|
> | qwen-2.5-7b        | 24.0 (1.0)     | 63.0 (0.016)   | **65.0** (0.273)   |
> | gemini-2.5-pro | 87.5 (1.0)     | 80.0 (0.016)   | **89.5** (0.669)   |
>
>
> [1] Wang, Chonghua, et al. "Ada-leval: Evaluating long-context llms with length-adaptable benchmarks." arXiv preprint arXiv:2404.06480 (2024).

---

> > ### Author Response · Authors · 2025-11-27
> >
> > **Q3: Zero-shot section (Sec. 5.5) lacks mechanism analysis. Authors transfer LaRA-trained policies to HELMET (HotpotQA/NQ) and report modest gains, but do not explain why a Comparison-trained policy improves HotpotQA or why a Location-trained policy improves NQ. Given that different tasks have vastly different retrieval needs, this positive result may be a coincidence.**
> >
> > Thank you for your insightful feedback. We agree with the reviewer that explaining why zero-shot transfer works is important. Most tasks in LaRA (such as Location) are single-hop QA tasks, where the answer can be derived from a single relevant passage. In contrast, the Comparison task in LaRA and HotpotQA in HELMET both require multi-hop reasoning, where multiple passages must be retrieved and combined to produce the correct answer (Line 321, 372).
> >
> > Our analysis in Section 5.3 shows that for multi-hop tasks, LDAR optimizes to retrieve a larger number of passages compared to other single-hop tasks (e.g., Location task) (Line 428). Based on this observed behavior, we want to highlight that our zero-shot mapping is structurally motivated.
> >
> > To further validate that the positive zero-shot results are not coincidental, we conducted an additional misaligned transfer experiment. Specifically, we evaluate **(1) Location-trained LDAR (single-hop) transferred to HotpotQA (multi-hop)**, and **(2) Comparison-trained LDAR (multi-hop) transferred to NQ (single-hop)**. As shown in the table below, misaligned transfers lead to degraded performance: Location → HotpotQA retrieves too few passages to support multi-hop reasoning, while Comparison → NQ retrieves too many passages, introducing additional distraction that harms performance. These results demonstrate that LDAR’s zero-shot transfer benefits arise from alignment between task structure, rather than coincidence.
> >
> > | Task aligned |            | HotpotQA           |            |                 | NQ                |                 |
> > |----------------|--------------------|--------------------|--------------------|-------------------|-------------------|-------------------|
> > |                | LC                 | RAG                | LDAR               | LC                | RAG               | LDAR              |
> > | qwen-3-4b        | 0.56 (1.0)         | 0.51 (0.019)       | **0.62** (0.536)   \|| 0.51 (1.0)        | 0.41 (0.021)      | **0.53** (0.486)  |
> > | gpt-4o-mini    | 0.64 (1.0)         | 0.65 (0.019)       | **0.76** (0.629)   \|| **0.59** (1.0)    | 0.52 (0.021)      | **0.59** (0.374)  |
> >
> > | Task misaligned |            | HotpotQA           |            |                 | NQ                |                 |
> > |----------------|--------------------|--------------------|--------------------|-------------------|-------------------|-------------------|
> > |                | LC                 | RAG                | LDAR               | LC                | RAG               | LDAR              |
> > | qwen-3-4b        | 0.56 (1.0)         | 0.51 (0.019)       | **0.60** (0.417)   \|| **0.51** (1.0)        | 0.41 (0.021)      | 0.50 (0.575)  |
> > | gpt-4o-mini    | 0.64 (1.0)         | 0.65 (0.019)       | **0.74** (0.493)   \|| **0.59** (1.0)    | 0.52 (0.021)      | 0.58 (0.424)  |

---

> > > ### Author Response · Authors · 2025-11-27
> > >
> > > **Q4: The paper employs a Transformer Encoder to process the sequence of similarity scores. This is a non-trivial architectural choice, implying that the relationships between scores are important. However, this is not justified with a baseline, such as a simpler Multi-Layer Perceptron (MLP), to prove this complexity is necessary.**
> > >
> > > **&**
> > >
> > > **Q5: The model uses "Periodic Embedding" to encode the raw scalar similarity scores. This is a specific and non-obvious design choice. The paper provides no justification or ablation study comparing this to simpler, standard alternatives (such as learnable embedding with truncation of sorted scores).**
> > >
> > > Thank you for your suggestion. As the number of passages associated with each query varies (Section 4.1), it produces similarity vectors of different lengths. We used Transformer encoder as it can handle variable-length sequences and is able to model relationships within the similarity distribution. For our MLP variant, we followed the common practice by summarizing each similarity vector into a fixed-size representation using simple pooling (i.e., taking the mean over similarity scores across passages), and feed this pooled representation into a stack of MLP layers [2]. The ablation below shows that LDAR with the transformer encoder shows significantly better performance compared to LDAR with MLP.
> > >
> > > We used periodic embeddings because they are highly effective for capturing fine-grained variation in continuous inputs [3]. As shown in the ablation below, LDAR with periodic embeddings achieves significantly better performance than LDAR using a standard learnable embedding.
> > >
> > > | LLM           | Architecture                               | location       | reasoning      | comp           | hallu          | average            |
> > > |---------------|--------------------------------------------|----------------|----------------|----------------|----------------|----------------|
> > > | llama-3.1-8b  | Transformer with periodic embedding (LDAR) | **77.6** (0.216)   | **59.6** (0.361)   | **41.5** (0.501)   | **73.3** (0.217)   | **63.0** (0.324)   |
> > > | llama-3.1-8b  | Transformer with learnable embedding      | 77.5 (0.359)   | 55.8 (0.158)   | 36.5 (0.353)   | 67.5 (0.359)   | 59.3 (0.307)   |
> > > | llama-3.1-8b  | MLP                                        | 71.4 (0.319)   | 54.5 (0.286)   | 34.1 (0.424)   | 70.1 (0.323)   | 57.5 (0.338)   |
> > > ||||||||
> > > | gpt-4o-mini   | Transformer with periodic embedding (LDAR) | **88.8** (0.397)   | **80.5** (0.254)   | **63.4** (0.613)   | **51.8** (0.397)   | **71.1** (0.415)   |
> > > | gpt-4o-mini   | Transformer with learnable embedding                        | 86.7 (0.520)   | 79.2 (0.285)   | 58.5 (0.600)   | 35.0 (0.521)   | 64.9 (0.481)   |
> > > | gpt-4o-mini   | MLP                                        | 84.6 (0.332)   | 77.9 (0.307)   | 56.0 (0.497)   | 29.8 (0.335)   | 63.3 (0.367)   |
> > >
> > > [2] Zaheer, Manzil, et al. "Deep sets." *Advances in neural information processing systems* 30 (2017).
> > >
> > > [3] Gorishniy, Yury, Ivan Rubachev, and Artem Babenko. "On embeddings for numerical features in tabular deep learning." Advances in Neural Information Processing Systems 35 (2022): 24991-25004.

---

> > > > ### Author Response · Authors · 2025-11-27
> > > >
> > > > **Q6: The paper's central claim is improved inference efficiency (i.e., lower token usage). However, it provides no information whatsoever about the training cost. The RL training loop, as formulated in Eq. (3), requires a forward pass of the (potentially massive) pretrained LLM to obtain the reward signal for every gradient step. This suggests the training process may be prohibitively expensive. The paper provides no data on the number of LLM calls required for convergence or the total wall-clock training time. This omission makes it impossible to assess the method's practical viability or cost-benefit trade-off.**
> > > >
> > > > Thank you for highlighting this important point. We would like to first introduce LDAR’s inference-time efficiency, as it directly influences LDAR’s overall training cost.
> > > >
> > > > LDAR introduces a small additional computation during inference because it performs a forward pass of the learned LDAR retrieval strategy $\pi_\theta$. To quantify this overhead, we measured the end-to-end inference time per example in Location task. Specifically, the inference time includes the time required to (1) compute the query and passage embeddings, (2) run the bridge model (e.g., reranker or LDAR retriever), and (3) process the selected passages through the prediction LLM to generate the final answer. Note as baseline methods distribute their computation differently across these three stages, their total inference cost varies accordingly.
> > > >
> > > > The table below summarizes the average per-example inference time across all methods using both open-source LLMs and closed-source LLMs. With open-source models, LC exhibits the highest latency because it forwards all retrieved passages to the prediction LLM. Methods such as RAG, RankZephyr, and BGM (which rely on text-based rerankers) also incur substantial overhead due to heavy cross-encoder computation. In contrast, LDAR employs a lightweight, text-free adaptive retrieval mechanism, selecting passages to minimize potential interference from distracting passages. As a result, LDAR achieves the fastest inference time among LC and all reranker baselines, while simultaneously achieving the best overall performance.
> > > >
> > > > In the closed-source setting, the prediction LLM is highly optimized for fast inference, making total inference time less sensitive to input length. Even under these conditions, LDAR remains faster than LC and all reranker baselines while also achieving better performance. These results demonstrate that LDAR provides distraction-aware and long-context-capability-aware retrieval in a computationally efficient manner at inference time.
> > > >
> > > > | Metric                | Top-1         | Top-5         | Top-10        | Top-25        | Top-50        | LC            | RAG           | Self-Route    | Adaptive-k    | BGM           | RankZephyr    | LDAR              |
> > > > |-----------------------|--------------|--------------|--------------|--------------|--------------|--------------|--------------|--------------|--------------|--------------|--------------|-------------------|
> > > > | Time (Open-source)    | 1.3 (0.55)   | 1.5 (0.73)   | 1.7 (0.91)   | 2.5 (0.99)   | 4.0 (1.31)   | 15.4 (5.17)  | 18.4 (0.86)  | 4.5 (5.93)   | 8.6 (11.70)  | 13.3 (1.14)  | 10.2 (1.07)  | 3.9 (1.61)       |
> > > > | Time (Closed-source)  | 3.9 (2.29)   | 4.7 (1.91)   | 5.2 (2.36)   | 5.8 (2.52)   | 5.9 (2.20)   | 8.2 (2.67)   | 22.6 (1.75)  | 9.0 (4.09)   | 6.5 (4.70)   | 17.8 (5.94)  | 13.6 (2.56)  | 8.0 (2.69)       |
> > > > ||||||||||||||
> > > > | Score (Open-source)   | 31.1 (0.01)  | 60.1 (0.03)  | 66.9 (0.05)  | 71.3 (0.13)  | 67.9 (0.27)  | 56.2 (1.00)  | 71.6 (0.03)  | 65.8 (0.19)  | 47.7 (0.40)  | 68.9 (0.02)  | 56.1 (0.03)  | **77.3** (0.21)  |
> > > > | Score (Closed-source) | 31.5 (0.01)  | 61.7 (0.03)  | 70.8 (0.05)  | 80.0 (0.13)  | 80.3 (0.27)  | 88.0 (1.00)  | 75.2 (0.03)  | 80.0 (0.28)  | 64.4 (0.40)  | 72.2 (0.02)  | 62.7 (0.03)  | **90.5** (0.50)  |

---

> > > > > ### Author Response · Authors · 2025-11-27
> > > > >
> > > > > Based on the analysis of LDAR’s inference-time efficiency, we now illustrate how the cost benefits accumulate over time when deploying the learned LDAR strategy. For each LDAR training epoch, we compute (1) the cumulative training cost up to that epoch and (2) the inference cost when applying LDAR policy learned until that epoch with a certain number of inference calls (10K, 100K, and 500K inference calls). We then plot the cumulative cost on the x-axis and the resulting performance on the y-axis. **Figure 9 in Appendix D.3 (top, bottom)** shows the results for using open-source and closed-source LLMs are prediction models, respectively. Because LDAR has lower inference-time overhead compared to LC, its cost advantage becomes increasingly pronounced as the number of deployments grows.
> > > > >
> > > > > Importantly, as LDAR trains, it learns to identify the most effective passage set that minimizes distraction. This not only improves performance but also **reduces token usage during both training and inference**. As shown in the center plot of Figure 9 (bottom), LDAR progressively selects fewer tokens over training epochs, thereby lowering the per-epoch training cost as well. CAG [4] is a cache-augmented generation method that speeds up inference by preloading documents and reusing a precomputed KV-cache. We included CAG as an additional baseline to compare inference-time efficiency with LDAR.
> > > > >
> > > > > Additionally, to provide a fair comparison in terms of API usage, we converted all training and inference costs into USD and visualized the results in **Figure 10 in Appendix D.3**. GPU hours are converted into USD based on the pricing of the cloud computing services we used, and API costs are computed according to the official pricing of OpenAI and Google. Under this USD-based metric, LDAR demonstrates greater cost efficiency than LC, particularly when closed-source LLMs are used as the prediction model, as the cost of processing long-context inputs is extremely high.
> > > > >
> > > > > [4] Chan, Brian J., et al. "Don't do rag: When cache-augmented generation is all you need for knowledge tasks." Companion Proceedings of the ACM on Web Conference 2025. 2025.

---

### Official Review · Reviewer_771K · 2025-11-01

**Soundness:** 3
**Presentation:** 2
**Contribution:** 3
**Rating:** 4
**Confidence:** 4

**Summary:**

The paper Beyond RAG vs. Long-Context: Learning Distraction-Aware Retrieval for Grounding LLMs studies the trade-off between retrieval-augmented generation (RAG), which may miss relevant information, and long-context models, which can include too much irrelevant content. It introduces LDAR (Learning Distraction-Aware Retrieval), a lightweight retrieval method that adaptively selects a contiguous range of passages from the ranked candidates based only on their similarity scores, achieving a balance between information coverage and distraction. LDAR uses a small transformer trained with policy gradients to predict how wide the retrieval range should be for each query, optimizing task performance according to the LLM’s context capacity. Experiments on multiple benchmarks and models show that LDAR achieves higher accuracy than both RAG and long-context approaches while using fewer tokens. The main contributions are a learning-based distraction-aware retrieval framework, the banded retrieval design that improves stability and generalization, and extensive validation demonstrating efficient grounding for LLMs.

**Strengths:**

1) The paper proposes a dynamic interval (band) method to select high-quality context from a passage sequence sorted by cosine similarity.

2) The motivation section analyzes in depth how different retrieval strategies, when applied over a cosine-sorted list, affect answer correctness, and it systematically frames the problem of adaptively balancing recall versus distraction.

3) The evaluation covers a reasonably broad set of settings on four LaRA tasks (location/reasoning/comparison/hallucination) and the long-context variants of HotpotQA/NQ from HELMET.

**Weaknesses:**

1) The analysis experiments for the method itself are not sufficient (see Questions).

2) The method appears sensitive to LLM size, which affects cross-model generalization and thus reduces transferability and robustness across heterogeneous systems.

3) There is a lack of concrete case studies: results are mainly aggregate metrics and curves, without step-by-step comparisons on representative examples.

**Questions:**

1) Do different retrievers induce different cosine-similarity distributions, and would this shift the behavior/performance of the adaptive retriever?

2) Using only accuracy as the reward seems limited. Since the token usage ratio is also reported, can the optimization include a cost term (e.g., inference/compute budget)?

3) In the visualization of retrieval intervals (Fig. 6), most learned bands overlap with the top-k region. For bands outside the top-k, can you provide concrete examples to intuitively explain why passages outside the top interval are selected/beneficial (or why the top interval is insufficient)?

4) Is selecting a continuous interval (band) inherently reasonable? Does it implicitly assume that high-quality passages are locally clustered in the cosine-sorted order? If possible, please add a fixed-length sliding-window experiment: slide a window of length (w) over the cosine-sorted sequence, evaluate accuracy for each window as the retrieved set, examine whether stable peak regions exist, and compare their alignment with the LDAR-learned band.

---

> ### Author Response · Authors · 2025-11-27
>
> We thank the reviewer for the thorough and constructive comments. We hope we can address your concerns below.
>
> **Q1: There is a lack of concrete case studies: results are mainly aggregate metrics and curves, without step-by-step comparisons on representative examples.**
>
> Thank you for your suggestion. **Figure 8 in Appendix D.2** provides a detailed case study of LDAR’s retrieval behavior, complementing the example shown in Figure 3 of the main paper.
>
> When the similarity distribution contains a clear high-similarity region, LDAR tends to focus narrowly on that region. In these cases, the useful passage tends to be located within the top rank passages as illustrated in Figure 8 (top).
>
> Conversely, when the overall similarity distribution is low in value and relatively flat in shape, LDAR expands its retrieval interval to ensure broader coverage, even at the risk of including more distracting passages. In these cases, the useful passage is often not located within the top rank passages as illustrated in Figure 8 (bottom), making a wider retrieval band necessary.
>
> Although LDAR does not receive any textual information (Line 181), this case study demonstrates that the shape of the similarity distribution is predictive of where useful passages tend to appear. LDAR learns these patterns and adapts its retrieval strategy accordingly.
>
> **Q2: Do different retrievers induce different cosine-similarity distributions, and would this shift the behavior/performance of the adaptive retriever?**
>
> Thank you for your insightful comment. Indeed, different retrievers produce different cosine-similarity distributions, as they learn different semantic spaces and therefore assign different similarity magnitudes and rankings to the same query–passage pairs. For an LDAR policy trained with one retriever to generalize to another, we found that a certain degree of scale alignment and rank alignment between the two retrievers is neceassary.
>
> To illustrate this, **Figure 11 (left) in Appendix D.4** compares the mean cosine-similarity values produced by bge-large-en-v1.5 and gte-large-en-v1.5 over 100 randomly sampled query–passage pairs from the Location task. Figure 11 (right) further shows how passages ranked by the bge-large-en-v1.5 map to their corresponding ranks in the gte-large-en-v1.5. These plots demonstrate that the two embedding models exhibit some amount of **scale alignment** (their similarity ranges are comparable) and **rank alignment** (higher-ranked passages in one model tend to remain relatively high-ranked in the other. Correlation: 0.7715). Because of this alignment, LDAR policies trained with bge-large-en-v1.5 can generalize to gte-large-en-v1.5 to a reasonable extent, and vice versa, as shown in the table below.
>
> | Location                | LC   | RAG  | LDAR|
> |-------------------------|------|------|-------------------|
> | bge-large-en-v1.5       | 69.3 (1.0) | 72.4 (0.026) | **77.6** (0.216)  |
> | gte-large-en-v1.5       | 69.3 (1.0) | 70.4 (0.022) | **74.4** (0.438)  |
> | bge → gte-large-en-v1.5 | 69.3 (1.0) | 70.4 (0.022) | **74.4** (0.297)  |
> | gte → bge-large-en-v1.5 | 69.3 (1.0) | 72.4 (0.026) | **75.5** (0.344)  |
>
> | Reasoning               | LC   | RAG  | LDAR|
> |-------------------------|------|------|-------------------|
> | bge-large-en-v1.5       | 50.6 (1.0) | 45.4 (0.024) | **59.6** (0.361)  |
> | gte-large-en-v1.5       | 50.6 (1.0) | 51.9 (0.021) | **57.1** (0.307)  |
> | bge → gte-large-en-v1.5 | 50.6 (1.0) | 51.9 (0.021) | **58.4** (0.252)  |
> | gte → bge-large-en-v1.5 | 50.6 (1.0) | 45.4 (0.024) | **54.5** (0.387)  |
>
> | Comp                    | LC   | RAG  | LDAR|
> |-------------------------|------|------|-------------------|
> | bge-large-en-v1.5       | 34.1 (1.0) | 19.5 (0.025) | **41.5** (0.501)  |
> | gte-large-en-v1.5       | 34.1 (1.0) | 24.3 (0.022) | **39.0** (0.536)  |
> | bge → gte-large-en-v1.5 | **34.1** (1.0) | 24.3 (0.022) | 26.8 (0.624)  |
> | gte → bge-large-en-v1.5 | **34.1** (1.0) | 19.5 (0025) | 24.9 (0.473)  |
>
> | Hallu                     | LC   | RAG  | LDAR|
> |-------------------------|------|------|---------------------|
> | bge-large-en-v1.5       | 58.4 (1.0) | **78.3** (0.026) | 73.3 (0.217)    |
> | gte-large-en-v1.5       | 58.4 (1.0) | **77.9** (0.022) | 70.1 (0.438)    |
> | bge → gte-large-en-v1.5 | 58.4 (1.0) | **77.9** (0.022) | 71.4 (0.311)    |
> | gte → bge-large-en-v1.5 | 58.4 (1.0) | **78.3** (0.026) | 68.8 (0.360)    |
>
> | Average                 | LC   | RAG  | LDAR|
> |-------------------------|------|------|-------------------|
> | bge-large-en-v1.5       | 53.1 (1.0) | 53.9 (0.025) | **63.0** (0.324)  |
> | gte-large-en-v1.5       | 53.1 (1.0) | 56.1 (0.022) | **60.1** (0.429)  |
> | bge → gte-large-en-v1.5 | 53.1 (1.0) | 56.1 (0.022) | **57.7** (0.371)  |
> | gte → bge-large-en-v1.5 | 53.1 (1.0) | 53.9 (0.025) | **55.9** (0.391)  |

---

> > ### Author Response · Authors · 2025-11-27
> >
> > **Q3: Using only accuracy as the reward seems limited. Since the token usage ratio is also reported, can the optimization include a cost term (e.g., inference/compute budget)?**
> >
> > Thank your for your insightful feedback. At first, we included a penalty term proportional to the number of retrieved tokens during training, just as the reviewer pointed out. However, as shown in **Figure 12 in Appendix D.5**, introducing such a cost term caused the optimization to collapse into a local optimum, where the policy retrieved too few passages and suffered a drop in accuracy.
> >
> > Including a cost term would have been necessary if performance increases monotonically with the number of retrieved passages. However, both prior work (e.g., the inverted-U findings in [1, 2]) and our own experiments show that performance peaks at an intermediate retrieval size and then decreases, forming an inverted-U relationship due to distraction from additional passages. Because the objective is not a linear trade-off but rather identifying the peak of this inverted-U, we found penalizing token usage during training often pushes the model away from the optimal region.
> >
> > Therefore, instead of imposing an explicit cost term, we focused on letting LDAR learn to retrieve passages near the performance peak of the inverted-U. Notably, even without a cost term, LDAR naturally avoids the high-distraction region and achieves both higher accuracy and lower token usage compared to the long-context baseline (Table 1).
> >
> > **Q4: The method appears sensitive to LLM size, which affects cross-model generalization and thus reduces transferability and robustness across heterogeneous systems.**
> >
> > Thank you for your comment. LDAR indeed achieves the best performance when it is optimized with the target LLM, which is consistent with the prior works on bridging retriever–LLM preference gaps (Section 2).
> >
> > Moreover, as shown in Table 1, we want to note that **all baseline methods** (including RAG, LC, BGM, Self-Route, Adaptive-k, and RankZephyr) **are also highly sensitive to LLM size**. This sensitivity is expected because long-context capability varies substantially across models. Within this landscape, LDAR consistently outperforms competing baselines in both open-source and closed-source settings, offering a meaningful and robust way to improve model performance given each LLM’s long-context capability.
> >
> > [1] Jin, Bowen, et al. "Long-context llms meet rag: Overcoming challenges for long inputs in rag." *arXiv preprint arXiv:2410.05983* (2024).
> >
> > [2] Leng, Quinn, et al. "Long context rag performance of large language models." *arXiv preprint arXiv:2411.03538* (2024).

---

> ### Author Response · Authors · 2025-11-27
>
> **Q5: In the visualization of retrieval intervals (Fig. 6), most learned bands overlap with the top-k region. For bands outside the top-k, can you provide concrete examples to intuitively explain why passages outside the top interval are selected/beneficial (or why the top interval is insufficient)?**
>
> Thank you for your thoughtful question. To better understand why LDAR sometimes selects passages outside the top-k region, we performed a clustering analysis over LDAR’s retrieval behavior on both an open-source model (qwen-2.5-7b) and a closed-source model (gemini-2.5-pro) **(Figure 13, 14 in Appendix D.6)**. We collected 100 similarity distributions and clustered them based on LDAR’s retrieval band value, resulting in two primary clusters (Cluster 0 and Cluster 1). The figure summarizes the mean similarity distribution for each cluster (Cluster 0, Cluster 1) along with LDAR’s average retrieval band ($q_L$, $q_U$).
>
> The visualization demonstrates that when the similarity distribution contains a relatively high-similarity region, LDAR concentrates its retrieval on that narrow interval (Cluster 0). Conversely, when similarity values are relatively low, LDAR expands the retrieval range to increase information coverage (Cluster 1). To specifically examine cases where LDAR avoids the top interval, we additionally isolated all instances where the learned upper quantile satisfies $q_U < 1.0$. These cases typically fall within Cluster 1, but we regrouped them separately to analyze this behavior in finer detail. As illustrated in the third plot, these distributions exhibit the lowest overall similarity values.
>
> To better understand why LDAR band shifts downward in such cases, we performed a fixed-width sliding-window experiment **(Figure 18 in Appendix D.8)**. For each cluster, we swept windows of size $w \in \\{20, 50, 100\\}$ across the similarity-ranked passages and measured accuracy using each window as the retrieved set. The sliding-window experiment demonstrates that when the similarity distribution contains a relatively high-similarity region (Cluster 0), narrower interval (w=20, 50) shows a better performance peak than wider interval (w=100).  Conversely, when similarity values are relatively low (Cluster 1), wider interval (w=100) shows a better performance peak. This behavior aligns with LDAR’s learned retrieval bands. Interestingly, for Cluster 1 we observe a secondary performance peak when a large window (w=100) covers the mid-quantile region. This suggests that, in low-similarity scenarios, informative passages are often dispersed across the middle ranks rather than concentrated at the very top. We believe such occasional peaks in the mid-quantile region made LDAR to occasionally shifts its retrieval band downward to select mid-similarity passages when the overall similarity distribution is low. The sliding-window experiment for the group $q_U < 1.0$ **(Figure 18 (bottom))** further shows that the highest-ranked passage can sometimes act as a distractor when the overall similarity distribution is low.
>
> By comparing LDAR’s behavior across open-source and closed-source LLMs **(Figure 13, 14)**, LDAR generally uses wider quantile bands when interacting with closed-source models, reflecting the fact that these models possess stronger long-context processing capabilities. Additionally, the frequency of non–top-interval retrieval (i.e., cases where $q_U < 1.0$) decreases notably for closed-source models. This indicates that closed-source LLMs are inherently more resilient to noisy or misleading passages, reducing the need for LDAR to shift its retrieval band downward to avoid distraction. Together, these observations highlight that LDAR internalizes and adapts to the long-context characteristics of the underlying LLM, producing retrieval behaviors that are both model-aware and capability-aligned.

---

> > ### Author Response · Authors · 2025-11-27
> >
> > **Q6: Is selecting a continuous interval (band) inherently reasonable? Does it implicitly assume that high-quality passages are locally clustered in the cosine-sorted order? If possible, please add a fixed-length sliding-window experiment: slide a window of length (w) over the cosine-sorted sequence, evaluate accuracy for each window as the retrieved set, examine whether stable peak regions exist, and compare their alignment with the LDAR-learned band.**
> >
> > Thank you for your insightful feedback. Retrieving a contiguous interval in similarity-sorted space is a widely adopted practice in RAG systems. For example, standard top-k retrieval (e.g., top-5) implicitly selects a continuous band from the highest-similarity region. This design is effective because semantically related passages tend to appear in nearby regions of the cosine-sorted ranking. Similarily, LDAR does not assume that high-quality passages are inherently clustered in the cosine-sorted order, but assume local neighborhoods in cosine similarity often correspond to semantically coherent groups.
> >
> > Following the reviewer’s request, we have done a fixed-length sliding-window experiment as introduced in Q5. **Figure 18 in Appendix D.8** shows that when the similarity distribution includes a relatively high-similarity region (Cluster 0), narrower windows (w = 20, 50) yield higher performance peaks than a wider window (w = 100). In contrast, when similarity values are generally low (Cluster 1), a wider window (w = 100) achieves a better peak. This pattern aligns with the retrieval bands learned by LDAR (Figure 18 (top)).

---

### Official Review · Reviewer_8eo8 · 2025-11-01

**Soundness:** 1
**Presentation:** 2
**Contribution:** 2
**Rating:** 2
**Confidence:** 5

**Summary:**

This work investigates the noisy information in the context of RAG, and introduces LDAR (learning distraction-aware retrieval) that learns to select passages to minimize interference from distracting passages. The training framework invites the target LLM in-loop to refine the retrieval model. The approach shows effective on the LaRA benchmark compared with several baseline settings on Llama 3.1 8B, Llama 3.2 3B, Qwen 2.5 7B, Qwen 3 4B, Mistral-Nemo-12B, GPT-4o, GPT-4o-mini, Gemini-2.5-pro, and Gemini-2.5-flash. Some interesting findings are shown, including the more capable models (e.g., GPT-4o and Gemini 2.5) perform well on the long-context setting.

**Strengths:**

* This work is well-motivated and easy to follow.

* The proposed method can improve the performances for a range of LLMs.

* The proposed method, which does not require the internal information such as attention values of the target LLM, is model-agnostic and can be applied with any LLMs including the closed ones such as GPT and Gemini.

**Weaknesses:**

* This work should be compared with other reranking methods, which are important related work to address. However, in the current manuscript, none of the reranking methods such as RankRAG is mentioned and compared.

* The experimental setting of the baseline methods is not comprehensive. The number of passages in RAG, LC, BMG, and Adaptive-k are not clear. Number of passages greatly impact the end performance. Without an analysis on the passage size, the contribution of the proposed method is hard to clarify.

* The proposed method adds computation cost during inference. An analysis of the runtime could be added.

**Questions:**

* What is the basic retrieval model (Top-k) in the experiments?

* What is the exact long-context setting in the experiments? The entire dataset or top-k with a very large k?

* From Table 2, LC looks powerful when the LLMs are capable. LC can be speed up with prompt cache (Cache-Augmented Generation). Could you compare the method with CAG in terms of accuracy and efficiency?

---

> ### Author Response · Authors · 2025-11-27
>
> Thank you for your valuable feedback. Please find our responses to your concerns below.
>
> **Q1: What is the exact long-context setting in the experiments? The entire dataset or top-k with a very large k?**
>
> Thanks for pointing this out. In Table 1, our long-context (LC) baseline follows the setting defined in the LaRA benchmark. That is, LC is evaluated by feeding the model the **entire document** (constructed to reach 32k or 128k tokens in total) rather than using top-k retrieval with a specific large k. As described in LaRA, supplying the full document is essential for (1) ensuring that the LC model always receives all potentially relevant information, and (2) faithfully measuring a model’s true long-context capability without truncation or heuristic filtering. Furthermore, the 128k context length setting is designed to match the context length limit of LLMs used in the experiments to evaluate their true long-context capability.
>
> **Q2: This work should be compared with other reranking methods, which are important related work to address. However, in the current manuscript, none of the reranking methods such as RankRAG is mentioned and compared.**
>
> Thank you for your suggestion. We want to kindly point out that our original manuscript already contains two reranking methods as baselines. Specifically, the “RAG” baseline applies a cross-encoder reranker (bge-reranker-large) to reorder the top-5 retrieved passages (Line 382). In addition, BGM [1] is included as a recent and competitive baseline that not only reranks retrieved passages, but also identifies and selects the necessary passages among the top-k candidates using a sequence-to-sequence relevance scoring module (Lines 383, 441). This behavior is conceptually aligned with the goal of RankRAG [2], which reorders and filters retrieved passages to isolate the most helpful subset.
>
> As the official code or model of RankRAG is not publicly available, we instead included RankZephyr [3], a publicly released listwise reranker that jointly processes multiple passages and outputs a global ranking, and is known to exhibit strong performance among available listwise rerankers [4]. RankZephyr jointly processes the top-50 similarity-retrieved passages and produces a global ranking over them, from which it selects the final top-5 passages. The performance gap between LDAR and reranker baselines (RAG, BGM, RankZephyr) (Table 1) highlights that LDAR’s distraction-aware and long-context-capability-aware retrieval provides benefits beyond what reranking can achieve.
>
> **Q3: What is the basic retrieval model (Top-k) in the experiments?**
>
> Thank you for your comment. As described in Line 380 of the manuscript, Top-k retrieves passages purely based on their embedding similarity to the query. Specifically, it selects the top-k highest-similarity passages without applying any reranking, filtering, or additional modeling. This differs from the “RAG” baseline, which first retrieves candidate passages and then applies an additional cross-encoder reranker to reorder them (Line 382).
>
> [1] Ke, Zixuan, et al. "Bridging the Preference Gap between Retrievers and LLMs." Proceedings of the 62nd Annual Meeting of the Association for Computational Linguistics (Volume 1: Long Papers). 2024.
>
> [2] Yu, Yue, et al. "Rankrag: Unifying context ranking with retrieval-augmented generation in llms." *Advances in Neural Information Processing Systems* 37 (2024): 121156-121184.
>
> [3] Pradeep, Ronak, Sahel Sharifymoghaddam, and Jimmy Lin. "RankZephyr: Effective and Robust Zero-Shot Listwise Reranking is a Breeze!." arXiv preprint arXiv:2312.02724 (2023).
>
> [4] Sharifymoghaddam, Sahel, et al. "Rankllm: A python package for reranking with llms." Proceedings of the 48th International ACM SIGIR Conference on Research and Development in Information Retrieval. 2025.

---

> > ### Author Response · Authors · 2025-11-27
> >
> > **Q4: The experimental setting of the baseline methods is not comprehensive. The number of passages in RAG, LC, BGM, and Adaptive-k are not clear. Number of passages greatly impact the end performance. Without an analysis on the passage size, the contribution of the proposed method is hard to clarify.**
> >
> > Thank you for your feedback. We want to clarify that the manuscript already reports the number of retrieved passages for all baselines. In Table 1, the values in parentheses denote the token-usage ratio relative to the Long-Context (LC) setting (Line 327). Because LC corresponds to retrieving all passages, this ratio provides a consistent and comparable measure of how many passages each method retrieves. For example, in the Location task under the 128k setting, LDAR retrieves 0.209 * 128,000 = 26,752 tokens, whereas LC retrieves the full 128,000 tokens (1.0 * 128,000 = 128,000).
> >
> > In addition, we followed the standard LaRA benchmark [5] chunking procedure, forming 600-token passages with 100-token overlap. As the size of each passage is equal in length, we can compute the number of passages directly from the information provided in Table 1. This results in approximately **64 passages** for the 32k context-length setting and **256 passages** for the 128k setting, which are used consistently across all methods. Since the reported token-usage ratio corresponds exactly to the fraction of passages retrieved, the number of passages retrieved by each baseline can be computed directly from the table.
> >
> > Across both 32k and 128k settings, LDAR achieves significantly higher performance compared to baselines, while using about half the token budget of the long-context approach with both open-sourced and closed-source LLMs. This demonstrates that LDAR’s improvements are not due to retrieving more passages, but rather to its distraction-aware and long-context-capability-aware passage selection.
> >
> > [5] Li, Kuan, et al. "LaRA: Benchmarking Retrieval-Augmented Generation and Long-Context LLMs--No Silver Bullet for LC or RAG Routing." arXiv preprint arXiv:2502.09977 (2025).

---

> > > ### Author Response · Authors · 2025-11-27
> > >
> > > **Q5: The proposed method adds computation cost during inference. An analysis of the runtime could be added.**
> > >
> > > Thank you for highlighting this important point. As noted by the reviewer, LDAR introduces a small additional computation during inference because it performs a forward pass of the learned LDAR retrieval strategy $\pi_\theta$. To quantify and compare this cost, we measured the end-to-end inference time per example in Location task. Specifically, the inference time includes the time required to (1) compute the query and passage embeddings, (2) run the bridge model (e.g., reranker or LDAR retriever), and (3) process the selected passages through the prediction LLM to generate the final answer. Note as baseline methods distribute their computation differently across these three stages, their total inference cost varies accordingly.
> > >
> > > The table below summarizes the average per-example inference time across all methods using both open-source LLMs and closed-source LLMs. With open-source models, LC exhibits the highest latency because it forwards all retrieved passages to the prediction LLM. Methods such as RAG, RankZephyr, and BGM (which rely on text-based rerankers) also incur substantial overhead due to heavy cross-encoder computation. In contrast, LDAR employs a lightweight, **text-free** adaptive retrieval mechanism, selecting passages to minimize potential interference from distracting passages. As a result, LDAR achieves **the fastest inference time among LC and all reranker baselines**, while simultaneously achieving the best overall performance.
> > >
> > > In the closed-source setting, the prediction LLM is highly optimized for fast inference, making total inference time less sensitive to input length. Even under these conditions, LDAR remains faster than LC and all reranker baselines while also achieving better performance. These results show that LDAR provides distraction-aware and long-context-capability-aware retrieval in a computationally efficient manner at inference time.
> > >
> > > | Metric                | Top-1         | Top-5         | Top-10        | Top-25        | Top-50        | LC            | RAG           | Self-Route    | Adaptive-k    | BGM           | RankZephyr    | LDAR              |
> > > |-----------------------|--------------|--------------|--------------|--------------|--------------|--------------|--------------|--------------|--------------|--------------|--------------|-------------------|
> > > | Time (Open-source)    | 1.3 (0.55)   | 1.5 (0.73)   | 1.7 (0.91)   | 2.5 (0.99)   | 4.0 (1.31)   | 15.4 (5.17)  | 18.4 (0.86)  | 4.5 (5.93)   | 8.6 (11.70)  | 13.3 (1.14)  | 10.2 (1.07)  | 3.9 (1.61)       |
> > > | Time (Closed-source)  | 3.9 (2.29)   | 4.7 (1.91)   | 5.2 (2.36)   | 5.8 (2.52)   | 5.9 (2.20)   | 8.2 (2.67)   | 22.6 (1.75)  | 9.0 (4.09)   | 6.5 (4.70)   | 17.8 (5.94)  | 13.6 (2.56)  | 8.0 (2.69)       |
> > > | Score (Open-source)   | 31.1 (0.01)  | 60.1 (0.03)  | 66.9 (0.05)  | 71.3 (0.13)  | 67.9 (0.27)  | 56.2 (1.00)  | 71.6 (0.03)  | 65.8 (0.19)  | 47.7 (0.40)  | 68.9 (0.02)  | 56.1 (0.03)  | **77.3** (0.21)  |
> > > | Score (Closed-source) | 31.5 (0.01)  | 61.7 (0.03)  | 70.8 (0.05)  | 80.0 (0.13)  | 80.3 (0.27)  | 88.0 (1.00)  | 75.2 (0.03)  | 80.0 (0.28)  | 64.4 (0.40)  | 72.2 (0.02)  | 62.7 (0.03)  | **90.5** (0.50)  |
> > >
> > > In addition, we include an analysis illustrating **how the cost benefits accumulate over time** when deploying the learned LDAR strategy. For each LDAR training epoch, we compute (1) the cumulative training cost up to that epoch and (2) the inference cost when applying LDAR policy learned until that epoch with a certain number of inference calls (10K, 100K, and 500K inference calls). We then plot the cumulative cost on the x-axis and the resulting performance on the y-axis. **Figure 9 (top, bottom) in Appendix D.3** shows the results for using open-source and closed-source LLMs as prediction models, respectively. Because LDAR has lower inference-time overhead compared to LC, its cost advantage becomes increasingly pronounced as the number of deployments grows.
> > >
> > > Importantly, as LDAR trains, it learns to identify the most effective passage set that minimizes distraction. This not only improves performance but also **reduces token usage during both training and inference**. As shown in the center plot of Figure 9 (bottom), LDAR progressively selects fewer tokens over training epochs, thereby lowering the per-epoch training cost as well.
> > >
> > > Additionally, to provide a fair comparison in terms of API usage, we converted all training and inference costs into USD and visualized the results **(Figure 10 in Appendix D.3)**. GPU hours are converted into USD based on the pricing of the cloud computing services we used, and API costs are computed according to the official pricing. Under this USD-based metric, LDAR demonstrates greater cost efficiency than LC, particularly when closed-source LLMs are used as the prediction model, as the cost of processing long-context inputs is extremely high.

---

> > > > ### Author Response · Authors · 2025-11-27
> > > >
> > > > **Q6: From Table 2, LC looks powerful when the LLMs are capable. LC can be speed up with prompt cache (Cache-Augmented Generation). Could you compare the method with CAG in terms of accuracy and efficiency?**
> > > >
> > > > Thank you for your feedback. CAG [6] operates by first computing a KV-cache over a combined set of documents \$D = \\{d_1, d_2, \ldots\\}\$ that corresponds to a set of queries $\\{q_1, q_2, \ldots\\}$. Once this large unified KV-cache is constructed, the model can process each instance efficiently by reusing the cached representations. In their experiments on HotPotQA, the largest-scale setting aggregates 64 documents (approximately 85k tokens) to build this cache.
> > > >
> > > > However, in long-context benchmarks such as LaRA, a single document associated with a single query already reaches the context-length limit of LLMs (which is 128k tokens in our experiments). Therefore, computing a KV-cache over multiple documents is infeasible under long-context setting, and **CAG effectively exhibits the same computational complexity as standard LC.**
> > > >
> > > > That said, the LaRA benchmark has a particular characteristic: each document is paired with multiple queries (e.g., 10 queries per document). This enables CAG to gain some benefit by caching individual documents and reusing their cached representations across queries that reference the same document. Accordingly, in our experiments, we implemented CAG by caching each document separately and applying the cache to all associated queries. We report results based on this fair and practically feasible adaptation of CAG for long-context evaluation. With such experimental settings, CAG achieved performance comparable to LC, but yields a 2.5× reduction in latency (15.41 seconds for LC vs. 4.31 seconds for CAG per inference on average).
> > > >
> > > > **Figure 9, 10 in Appendix D.3** compare LDAR, LC, and CAG under two cost metrics (GPU hours and USD). Note that we did not apply CAG when using closed-source LLMs as the prediction model, since constructing the required KV-cache is not feasible in this setting. While CAG shows some efficiency gains over LC, LDAR achieves both higher performance and greater cost efficiency as the number of deployments grows.
> > > >
> > > > [6] Chan, Brian J., et al. "Don't do rag: When cache-augmented generation is all you need for knowledge tasks." Companion Proceedings of the ACM on Web Conference 2025. 2025.

---

### Author Response · Authors · 2025-12-02
**Summary of Revision**

We sincerely thank all reviewers for their valuable feedback and thoughtful comments. We are glad reviewers found our findings interesting (`8eo8`), motivation solid (`8eo8`, `771K`, `VnKc`), experiments comprehensive (`771K`, `brC4`, `VnKc`), and our paper addressing a significant problem (`brC4`, `VnKc`).

In response to all received comments, we summarize the major revisions below, listed in the order of their appearance in the revised manuscript:

- **Clarifying the long-context setting** (`8eo8`): In `Section 5.2`, we added some details clarifying the long-context setting in Line 382.

- **Clarifying the reranker baselines** (`8eo8`): In `Section 5.2` and `Section 5.4`, we added some details clarifying that our original manuscript already contains two strong reranker baselines. Additionally, we included a listwise reranker (RankZephyr) as one of our baselines (Table 1).

- **Clarifying the zero-shot evaluation setting** (`brC4`): In `Section 5.5`, we added some details about the zero-shot transfer setting for clarification. `Appendix D.10` provides an additional task-misaligned transfer experiment.

- **Training/Inference Cost efficiency analysis** (`8eo8`, `brC4`): `Section 5.6` and `Appendix D.3` provide a detailed analysis of LDAR’s cost–benefit trade-off in both training and inference.

- **Clarifying the number of passages retrieved** (`8eo8`, `VnKc`): In `Appendix C.3`, we added details about the text chunking procedure used and clarified that the number of retrieved passages can be directly computed from the information provided in Table 1.

- **Cross-Embedder Generalization** (`771K`): `Appendix D.4` analyzes how and when LDAR strategy trained with one embedder can generalize to another.

- **Optimizing LDAR with a cost term** (`771K`): `Appendix D.5` provides a detailed analysis of how incorporating a cost term influences LDAR during training, and clarifies why LDAR is optimized solely on the evaluation signal.

- **Analysis of LDAR’s learned retrieval strategy** (`771K`, `VnKc`): `Appendix D.6` analyzes LDAR’s learned behavior through clustering analyses and fixed-width sliding-window experiments. In addition, Figure 8 in `Appendix D.2` provides a concrete case study of LDAR’s learned retrieval strategy for better interpretability.

- **Zero-shot Generalization analysis** (`brC4`): `Appendix D.2 (Figure 8)` and `Appendix D.7` explain how and when does LDAR succeeds in zero-shot generalization. We also included an additional experiment that zero-shot evaluates LDAR on a retrieval task built from StackOverflow question–answer pairs (Table 8).

- **Ablation study on architectural choices** (`brC4`): `Appendix D.9` presents ablation studies examining LDAR’s architectural components (Transformer encoder, periodic embedding).

We sincerely appreciate the reviewers' time and effort in providing constructive feedback and hope that the revised manuscript satisfactorily addresses all major concerns.

---

### Meta-Review · Area_Chair_pvh9 · 2026-01-02

**Summary:**

The paper introduced LDAR framework to address the trade-off between RAG and long-context models. LDAR employs a specialized transformer trained with policy gradients to analyze cosine similarity distributions and adaptively select passages to balance necessary information coverage with minimal distracting content. By incorporating the target LLM into the training loop to provide reward signals based on answer correctness, the method dynamically optimizes the retrieval range for the specific model capabilities. Extensive experiments show that LDAR achieves higher accuracy and token efficiency than baseline methods.

**Reviewer Concerns:**

Many reviewers raised the concern on the paper writing and experimental setting clarity. The authors addressed those comments during the rebuttal period (by revised paper as well). Some reviewers raised the concern on the computational complexity and the authors provided the quantitative results during the rebuttal period. Some overfitting issues are also raised but the authors addressed them successfully with additional quantitative experiments.

**Reviewer Scores:**

Initial scores are distributed from 2 to 8 (2,4,6,8). Unfortunately, no reviewers participated on the discussion during the rebuttal period. Thus, AC carefully checked all the responses provided by the authors. Especially, for the concerns raised by the reviewer (with score 2), those concerns are well addressed by the authors (e.g., experimental setting clarifications, additional baselines, as well as computational complexity). Furthermore, the concerns raised by another reviewer (with score 4) also resolved during the rebuttals (e.g., by providing additional experiments and analyses). Therefore, AC decided to accept this paper.

---

### Decision · Program_Chairs · 2026-01-26

Accept (Poster)